# Spen links RNA-mediated endogenous retrovirus silencing and X chromosome inactivation

Ava C Carter[1†], Jin Xu[1†], Meagan Y Nakamoto[2‡], Yuning Wei[1‡], Brian J Zarnegar[3], Quanming Shi[1], James P Broughton[1], Ryan C Ransom[4], Ankit Salhotra[4], Surya D Nagaraja[5], Rui Li[1], Diana R Dou[1], Kathryn E Yost[1], Seung-Woo Cho[1], Anil Mistry[6], Michael T Longaker[4,5], Paul A Khavari[3], Robert T Batey[2], Deborah S Wuttke[2], Howard Y Chang[1,3,7]*

[1]Center for Personal Dynamic Regulomes, Stanford University, Stanford, United States; [2]Department of Biochemistry, University of Colorado, Boulder, United States; [3]Department of Dermatology, Stanford University School of Medicine, Stanford, United States; [4]Department of Surgery, Division of Plastic and Reconstructive Surgery, Stanford University School of Medicine, Stanford, United States; [5]Institute for Stem Cell Biology and Regenerative Medicine, Stanford University, Stanford, United States; [6]Novartis Institute for Biomedical Research, Cambridge, United States; [7]Howard Hughes Medical Institute, Stanford University, Stanford, United States

*For correspondence:
howchang@stanford.edu

[†]These authors contributed equally to this work
[‡]These authors also contributed equally to this work

**Abstract** The *Xist* lncRNA mediates X chromosome inactivation (XCI). Here we show that Spen, an *Xist*-binding repressor protein essential for XCI , binds to ancient retroviral RNA, performing a surveillance role to recruit chromatin silencing machinery to these parasitic loci. Spen loss activates a subset of endogenous retroviral (ERV) elements in mouse embryonic stem cells, with gain of chromatin accessibility, active histone modifications, and *ERV* RNA transcription. Spen binds directly to *ERV* RNAs that show structural similarity to the A-repeat of *Xist*, a region critical for *Xist*-mediated gene silencing. *ERV* RNA and *Xist* A-repeat bind the RRM domains of Spen in a competitive manner. Insertion of an ERV into an A-repeat deficient Xist rescues binding of *Xist* RNA to Spen and results in strictly local gene silencing in *cis*. These results suggest that *Xist* may coopt transposable element RNA-protein interactions to repurpose powerful antiviral chromatin silencing machinery for sex chromosome dosage compensation.

## Introduction

*Xist* is a 17 kb long noncoding RNA that acts through specific interactions between its distinct RNA domains and nuclear effector proteins. The *Xist* RNA-associated protein complex was identified in 2015 using both genetic and affinity-based methods, and consists of multiple pleiotropic proteins, many of which are highly conserved throughout evolution and act on chromatin structure and gene regulation in myriad systems (*Augui et al., 2011*; *Chu et al., 2015*; *McHugh et al., 2015*; *Minajigi et al., 2015*; *Monfort et al., 2015*; *Moindrot et al., 2015*). This suggests that *Xist* evolved the ability to bind these proteins in the eutherian mammals, coopting those which evolved initially to perform other epigenetic functions. *Xist* evolved in the eutherian clade through exaptation of a combination of coding genes that were pseudogenized, as well as transposable elements (TEs) that inserted into this locus. *Xist* contains six tandem repeat regions (A-F), all of which show sequence similarity to TEs, suggesting they arose from eutheria-specific TE insertions (*Elisaphenko et al.,*

**eLife digest** The genetic material inside cells is often packaged into thread-like structures called chromosomes. In humans, mice and other mammals, a pair of sex chromosomes determines the genetic or chromosomal sex of each individual. Those who inherit two "X" chromosomes are said to be chromosomally female, while chromosomal males have one "X" and one "Y" chromosome. This means females have twice as many copies of genes on the X chromosome as a male does, which turns out to be double the number that the body needs.

To solve this problem, mammals have developed a strategy known as dosage compensation. The second X chromosome in females becomes "silent": its DNA remains unchanged, but none of the genes are active. A long noncoding RNA molecule called *Xist* is responsible for switching off the extra X genes in female cells. It does this by coating the entirety of the second X chromosome.

Normally, RNA molecules transmit the coded instructions in genes to the cellular machinery that manufactures proteins. "Noncoding" RNAs like *Xist*, however, are RNAs that have taken on different jobs inside the cell. Researchers believe that the ancestral Xist gene may have once encoded a protein but changed over time to produce only a noncoding RNA. Carter, Xu et al. therefore set out to find out how exactly this might have happened, and also how *Xist* might have acquired its ability to switch genes off.

Initial experiments used mouse cells grown in the laboratory, in which a protein called Spen was deleted. Spen is known to help *Xist* silence the X chromosome. In female cells lacking Spen, the second X chromosome remained active. Other chromosomes in male and female cells also had stretches of DNA that became active upon Spen's removal. These DNA sequences, termed endogenous retroviruses, were remnants of ancestral viral infections. In other words, Spen normally acted as an antiviral defense.

Analysis of genetic sequences showed that Spen recognized endogenous retrovirus sequences resembling a key region in *Xist*, a region which was needed for *Xist* to work properly. Inserting fragments of endogenous retroviruses into a defective version of *Xist* lacking this region also partially restored its ability to inactivate genes, suggesting that X chromosome silencing might work by hijacking cellular defenses against viruses. That is, female cells essentially 'pretend' there is a viral infection on the second X chromosome by coating it with *Xist* (which mimics endogenous retroviruses), thus directing Spen to shut it down.

This research is an important step towards understanding how female cells carry out dosage compensation in mammals. More broadly, it sheds new light on how ancient viruses may have shaped the evolution of noncoding RNAs in the human genome.

*2008*). One of these is the A-repeat, which is essential for gene silencing. When this ~500 bp region is deleted, *Xist* RNA coats the X chromosome, but silencing and reorganization of the X does not follow (*Wutz et al., 2002*; *Giorgetti et al., 2016*). The A-repeat sequence is thought to derive from the insertion and duplication of an endogenous retrovirus (ERV), a class of TEs present in many copies throughout the genome (*Elisaphenko et al., 2008*). In general, lncRNAs are not well-conserved compared to protein-coding genes but are enriched for TE content, suggesting they may be able to rapidly evolve functional domains by exapting protein- and nucleic acid-binding activity from entire TEs that colonize their loci (*Johnson and Guigó, 2014*; *Kelley and Rinn, 2012*). Understanding how the *Xist* RNA sequence was evolutionarily stitched together from these existing building blocks to gain protein-binding potential is of great interest towards understanding dosage compensation and lncRNA-mediated gene regulation genome-wide.

Spen (also known as SHARP, MINT) is a ~ 400 kDa *Xist* RNA binding protein (RBP) that contains four canonical RNA binding domains, as well as a SPOC domain to facilitate protein-protein interactions. Spen is a co-repressor that binds to several chromatin remodeling complexes, including histone deacetylases (HDACs), and the NuRD complex (*McHugh et al., 2015*; *Shi et al., 2001*). Though now recognized for its central role in the eutherian-specific XCI process, Spen is an ancient protein that plays roles in gene repression during development in species including *Drosophila* and *Arabidopsis*, in addition to mice and humans (*Shi et al., 2001*; *Yabe et al., 2007*; *Kuang et al., 2000*; *Tsuji et al., 2007*; *Bäurle et al., 2007*). Spen binds directly to the A-repeat of *Xist* RNA and

*Spen* inactivation abrogates silencing of multiple X-linked genes, suggesting that the RNA-protein interaction between the A-repeat and Spen is an early and essential step in XCI (*Chu et al., 2015*; *McHugh et al., 2015*; *Monfort et al., 2015*; *Lu et al., 2016*).

## Results

To test the effect of Spen loss on gene regulation and chromosome accessibility during XCI and genome-wide during development, we performed ATAC-seq (*Buenrostro et al., 2013*) in haploid mouse embryonic stem cells (mESCs) harboring a doxycycline-inducible *Xist* transgene either in the wild-type (WT) context or with a full deletion of *Spen* (*Spen* KO) (*Monfort et al., 2015*; *Figure 1a*). Following 48 hr of *Xist* induction, WT cells demonstrated loss of chromatin accessibility at the majority of loci on the X chromosome (*Figure 1b–c*; *Figure 1—figure supplement 1a,b*). In two independent *Spen* KO mESC clones, we found no X chromosome site that is reproducibly silenced upon *Xist* induction, suggesting that Spen is absolutely required for gene silencing at the level of chromatin accessibility (*Figure 1—figure supplement 1b*). This complete failure of XCI, combined with Spen's direct binding to the A-repeat region (*Chu et al., 2015*; *McHugh et al., 2015*; *Monfort et al., 2015*), confirms that Spen's recruitment to the inactivating X chromosome is early and essential for XCI (*Dossin et al., 2020*; *Nesterova et al., 2019*).

Despite recent focus on Spen's role in XCI, Spen is an ancient protein that is known to act as an important RNA-protein scaffold in developmental processes in many species (*Shi et al., 2001*; *Yabe et al., 2007*; *Kuang et al., 2000*; *Tsuji et al., 2007*; *Bäurle et al., 2007*). Thus, how *Xist* evolved the ability to recruit Spen is of great interest. We hypothesized that understanding Spen's role in autosomal gene regulation and its target specificity might lend clues into how Spen was exapted to participate in the Eutherian-specific process of XCI (*Figure 1a*). Comparison of ATAC-seq data in WT and *Spen* KO mESCs revealed 288 sites on the autosomes that gain accessibility in two clones of *Spen* KO mESCs, compared to only 147 sites that lose accessibility (*Figure 1d*). This observation is consistent with chromatin de-repression in the absence of Spen's repressive function. The DNA elements that were more accessible in *Spen* KO mESCs are almost all distal to transcriptional start sites (TSS), in contrast to unchanging peaks and those that are less accessible, which encompassed both promoters and distal sites (*Figure 1—figure supplement 1c*). This indicated that DNA elements regulated by Spen on the autosomes are not found at gene promoters, but rather are found in heterochromatic, gene-poor regions.

To better understand Spen's autosomal targets, we performed genome ontology enrichment for the sites that are de-repressed in *Spen* KO mESCs using HOMER. While the set of sites confidently gaining accessibility in both *Spen* KO clones is relatively small, it showed a striking enrichment for TE-derived long terminal repeat (LTR) elements (*Figure 1e*; *Figure 1—figure supplement 1d*). When we looked more closely at subsets of these annotations, we found an enrichment specifically for endogenous retrovirus K (ERVK) TEs in these sites, which is reproduced in two independent *Spen* KO clones (*Figure 1f*). These ERVKs are enriched specifically for LTR elements in the RLTR13, RLTR9, and early transposon (ETn) families (*Figure 1f*). ERV-derived sequences in the genome have recently been shown to play important roles in genome regulation in embryonic cell types, serving as binding sites for transcription factors, indicating that Spen loss may activate ectopic regulatory regions (*Bourque et al., 2008*; *Grow et al., 2015*; *Macfarlan et al., 2012*). Because sequencing reads coming from TEs in the genome do not map uniquely, it is difficult to accurately quantify reads coming from a given ERV subfamily while also mapping them to the correct element. We thus mapped ATAC-seq reads to the genome, while either keeping only uniquely mapping reads, or allowing multi-mapping reads to randomly map to one location. Both of these methods revealed the same trend, demonstrating the de-repression of a subset of ERVK elements in *Spen* KO mESCs (*Figure 1g*).

The observation that TE-derived elements were activated in *Spen* KO mESCs was intriguing, given that the A-repeat region of Xist is itself believed to be derived from an ancient TE insertion (*Elisaphenko et al., 2008*). It has been posited that TEs, upon insertion into pseudogenes or noncoding loci may contribute functional protein-binding domains to noncoding RNAs (*Johnson and Guigó, 2014*). Indeed, noncoding RNAs, including Xist, are enriched for TE content (*Kelley and Rinn, 2012*). Furthermore, it is known that in a very distantly related species, the model plant *Arabidopsis*, the Spen homologs *fca* and *fpa* bind to and regulate transcription of TEs in the genome

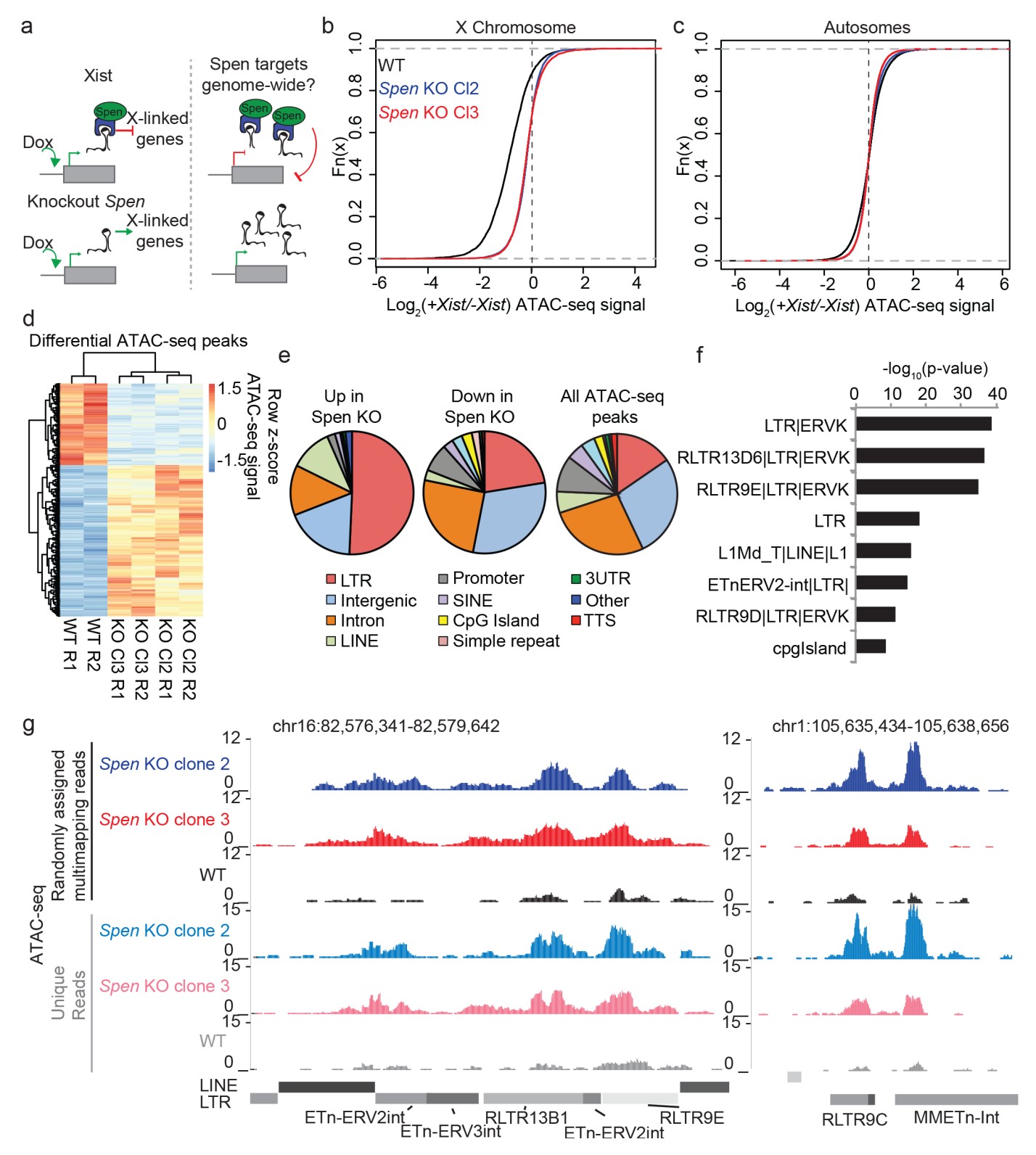

**Figure 1.** *Spen* knockout blocks XCI chromosome-wide and leads to derepression of autosomal ERVs. (a) Diagram of experimental set up. *Spen* KO mESCs are cultured +/- Doxycycline for 48 hr to induce *Xist* expression and then ATAC-seq is performed. We ask which sites are de-repressed in *Spen* KO compared to WT cells both during XCI and on the autosomes. (b) Log$_2$ ratio of ATAC-seq signal in +Dox and –Dox samples at all ATAC-seq peaks on the X chromosome in WT and *Spen* KO mESCs. Two independent *Spen* KO mESC clones are shown. (c) Same as in b for all autosomal peaks. (d) *Figure 1 continued on next page*

*Figure 1 continued*

Heatmap showing ATAC-seq signal for all differential peaks between WT and *Spen* KO mESCs. Values depict the z-scored value for normalized ATAC-seq reads within each peak for WT, *Spen* KO Cl2, and *Spen* KO Cl3 cells (with two replicates each). (e) Pie charts depicting the proportion of ATAC-seq peaks that cover each genomic region annotation. Annotations are derived from HOMER. (f) Genome Ontology enrichment (by HOMER) for all sites gaining accessibility in *Spen* KO mESCs compared to WT (*n* = 288). Plotted is the $-\log_{10}$ p-value for enrichment. *P* values (one-sided binomial) are corrected for multiple hypothesis testing using the Benjamini-Hochberg correction. (g) Two example ERVK regions on chromosome 16 (left) and chromosome 1 (right) showing a gain in chromatin accessibility specifically in two clones of *Spen* KO mESCs. Shown are tracks of uniquely mapping reads (top) and tracks where multimapping reads are randomly assigned to one location (bottom).

The online version of this article includes the following figure supplement(s) for figure 1:

**Figure supplement 1.** Spen is required for XCI chromosome-wide.

(*Bäurle et al., 2007*). Therefore, Spen's ability to regulate a subset of ERVK families may explain Spen's ability to recognize the A-repeat and interact directly with it.

In mESCs, the majority of TEs, including ERVKs, are silenced by histone H3 lysine nine trimethylation (H3K9me3). When members of the TE silencing machinery, such as Kap1 and Setdb1, are inactivated, H3K9me3 is lost, and these TE insertions can be expressed and function as ectopic promoters and enhancers (*Maksakova and Mager, 2005*; *Rowe et al., 2010*; *Karimi et al., 2011*; *Liu et al., 2014*). We hypothesized that in the absence of Spen and its protein partners, these ERVKs would lose H3K9me3 and gain histone modifications associated with active gene expression (*Żylicz et al., 2019*; *McHugh et al., 2015*; *Shi et al., 2001*).

Indeed, ChIP-seq experiments revealed that DNA elements gaining accessibility in *Spen* KO showed a dramatic loss of H3K9me3 and a gain of both histone H3 lysine 27 acetylation (H3K27Ac) and histone H3 lysine four trimethylation (H3K4me3) marks (*Figure 2a,b*; *Figure 2—figure supplement 1a–c*). Conversely, H3K9me3 peaks that are lost in *Spen* KO mESCs are enriched for ERVKs, and sites that gain the enhancer mark H3K27Ac are enriched for ERVKs including ETn elements (*Figure 2c,d*; *Figure 2—figure supplement 1d–f*). This demonstrates that at the level of chromatin modifications, a subset of ERVK elements are de-repressed when Spen is lost in mESCs. To test whether these ERVK elements are upregulated at the level of transcription, we performed RNA-seq in WT and *Spen* KO mESCs (*Figure 2—figure supplement 1a,e*; *Figure 2—figure supplement 2a–c*). At the RNA level, the ERVK elements that gain accessibility also have higher expression in *Spen* KO mESCs (*Figure 2b,e*; *Figure 2—figure supplement 1a*). Genome-wide there is a very small increase in expression of all ERVKs, particularly the class of ETnERV2s, indicating that only a subset of these families are affected by the loss of Spen in mESCs (*Figure 2f,g*). This is true using our multimapping read assignment strategy, as well as a published method for TE mapping, MMR (*Kahles et al., 2016*; *Figure 2—figure supplement 1g–i*).

Retrotransposons are parasitic elements whose insertion and propagation in cells depletes cellular resources and disrupts endogenous gene expression. In order for the cell to handle these parasitic elements, pathways have evolved to shut down these transposons at the level of the DNA locus and the RNA transcript. ERVK insertions that are more similar to the original TE sequence pose more of a threat to cells, as they are more likely to be mobile and able to replicate. Thus, we compared the sequence diversity of the ERVKs upregulated in S*pen* KO to those not upregulated in *Spen* KO, and found that those repressed by Spen have diverged less than those that are not repressed by Spen (*Figure 2h*). The lower diversity indicates that Spen targets more recent ERVK insertions. Furthermore, we found that genes involved in the innate cellular immunity pathway, which responds to retroviral RNA presence, are modestly upregulated in *Spen* KO mESCs (*Figure 2—figure supplement 2d*). These data suggest that Spen represses young, more intact ERVKs, and when Spen is inactivated, the cell responds to the presence of those RNA transcripts to attempt to silence them.

TE insertions are enriched for transcription factor binding motifs, and it has been suggested that insertion of many elements of the same family within the genome may have led to the evolution of transcription factor-based regulatory networks (*Bourque et al., 2008*; *Sundaram et al., 2014*; *Sundaram et al., 2017*). ERVK, and more specifically ETn, elements are enriched for binding sites for Oct4, one of the most critical transcription factors for regulation of the pluripotent state (*Bourque et al., 2008*; *Schöler, 1991*). Oct4 plays a critical role in balancing the expression of proself renewal and pro-differentiation genes in mESCs. We find that the Oct4 binding motif is enriched

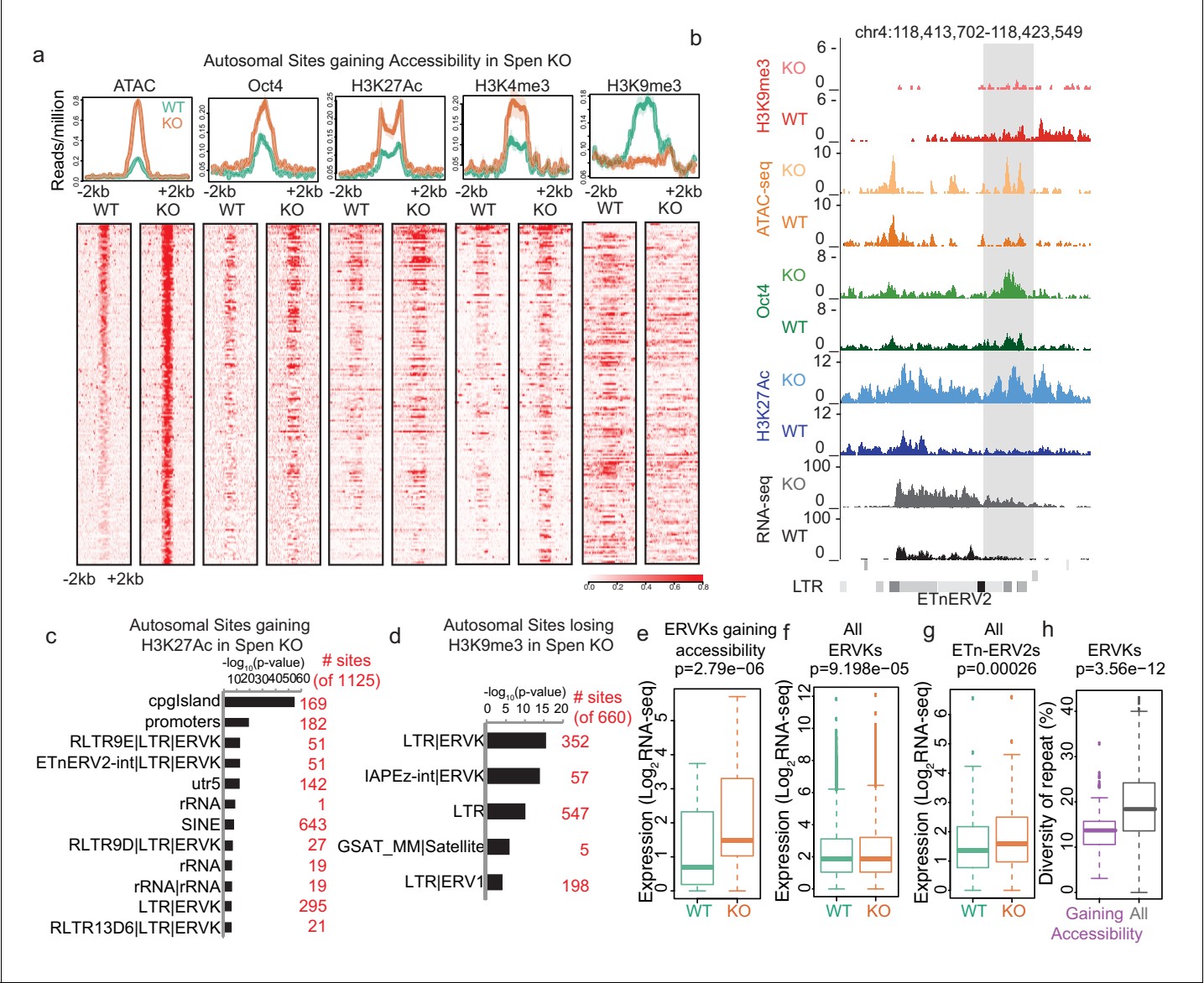

**Figure 2.** *Spen* knockout leads to gain of histone acetylation and loss of H3K9me3-mediated repression at ERVKs. (a) Average diagrams and heatmaps showing the read count per million mapped reads for ATAC-seq and ChIP-seq plotted over all ATAC-seq peaks that gain accessibility in *Spen* KO mESCs (*n* = 288) (+ / - 2 kb). From left to right are ATAC-seq, Oct4 ChIP-seq, H2K27Ac ChIP-seq, H3K4me3 ChIP-seq, and H3K9me3 ChIP-seq. Data is shown for WT mESCs (green) and *Spen* KO clone 2 mESCs (orange). For the heatmaps, each row represents one peak. (b) An example of an ETnERV2 element on chromosome four that shows a gain of chromatin accessibility, RNA expression, H3K27Ac ChIP-seq, Oct4 ChIP-seq, and a loss of H3K9me3 ChIP-seq in *Spen* KO mESCs compared to WT. (c) Genome Ontology enrichment (by HOMER) for all sites gaining H3K27Ac ChIP-seq signal in *Spen* KO mESCs compared to WT (*n* = 1125 sites). Plotted is the −log$_{10}$ p-value for enrichment. *P* values (one-sided binomial) are corrected using the Benjamini-Hochberg correction. (d) Genome Ontology enrichment (by HOMER) for all sites losing H3K9me3 ChIP-seq signal in *Spen* KO mESCs compared to WT (*n* = 660 sites). Plotted is the −log$_{10}$ p-value for enrichment. *P* values (one-sided binomial) are corrected using the Benjamini-Hochberg correction. (e) Expression of ERVKs gaining accessibility in *Spen* KO mESCs in WT mESCs (green) and *Spen* KO mESCs (orange). Shown is the log$_2$ transformed normalized RNA-seq read counts. For all boxplots, the thick line represents the median, while the box gives the IQR. *P* values are calculated using a two-tailed T test. (f) Same as in d for all ERVKs, genome wide. (g) Same as in e for all ETn-ERV2s. (h) Diversity of repeat for all ERVK elements that gain accessibility in *Spen* KO mESCs (purple) and all ERVK elements (gray). Percent diversity represents the percentage of the sequence that has diverged from the ancestral TE insertion. Percentages come from RepeatMasker.

The online version of this article includes the following figure supplement(s) for figure 2:

**Figure supplement 1.** *Spen* KO leads to dysregulation of ERVs.

**Figure supplement 2.** RNA-seq reveals downregulation of developmental genes in *Spen* KO mESCs.

**Figure supplement 3.** *Spen* KO mESCs fail to differentiate from the pluripotent state.

at DNA elements gaining accessibility in *Spen* KO mESCs, and Oct4 occupancy increases at these loci as shown by ChIP-seq (*Figure 2a*). The ability of this core stemness factor to bind to these TE elements suggests that their misregulation in *Spen* KO mESCs may affect the self-renewal or differentiation programs within these cells. GO term analysis revealed that genes downregulated in *Spen* KO mESCs are enriched for early developmental terms, indicating that these cells may be deficient in early lineage commitment (*Figure 2—figure supplement 2e–f*). We found that *Spen* KO mESCs downregulated specific genes associated with early differentiation and upregulated genes associated with self-renewal (*Figure 2—figure supplement 3a*).

Because the balance in the pluripotency transcriptional network is disrupted in *Spen* KO mESCs, we hypothesized that differentiation may be blocked. To test whether *Spen* KO mESCs can spontaneously differentiate when deprived of self-renewal signals, we removed leukemia inhibitory factor (LIF) from the media and allowed cells to grow for 6 days. After 6 days, WT mESCs downregulated pluripotency factors such as Oct4 and expressed early differentiation markers such as Nestin. In contrast, *Spen* KO mESCs maintained Oct4 expression and mESC morphology despite LIF removal (*Figure 2—figure supplement 3b,c*). We then tested whether *Spen* KO mESCs can be directed to differentiate toward the neurectoderm fate, by driving differentiation toward neural progenitor cells. After 14 days of differentiation, WT cells express Nestin and have NPC morphology while *Spen* KO mESCs do not (*Figure 2—figure supplement 3d*). Furthermore, we observed massive cell death during *Spen* KO mESC differentiation, finding that by day 10 of directed differentiation, 100% of cells were no longer viable (*Figure 2—figure supplement 3e*). These results support the results from RNA-seq that suggest that Spen loss leads to a failure to differentiate past the pluripotent state.

We next asked whether Spen represses ERVK loci directly or indirectly. Though we observe changes in chromatin accessibility and covalent histone modifications at these loci when Spen is knocked out, Spen is an RBP that does not bind to chromatin directly. We hypothesized that Spen may recognize *ERVK* RNAs transcribed at these TE loci and recruit chromatin silencing machinery to them, performing a surveillance role against aberrant transcription of these parasitic elements. Thus if Spen is regulating these ERVK loci directly, we would expect that regulation to be RNA-dependent (*Figure 3a*).

To test this, we performed infrared crosslinking immunoprecipitation followed by sequencing (irCLIP-seq), which allows for identification of direct RBP binding sites on RNA (*Zarnegar et al., 2016*). Because of Spen's large size (~400 kDa) and the lack of highly specific antibodies to it (*Figure 3—figure supplement 1a,b*), we expressed a FLAG-tagged version of the Spen RNA binding domains (RRM2-4) for irCLIP-seq. We also compared irCLIP-seq data performed using an antibody against the endogenous, full-length Spen protein, despite the higher level of non-specific signal in those experiments (*Figure 3—figure supplement 1a,b*; *Lu et al., 2019*). RRMs 2–4 were shown to be bona fide RNA binding domains, with RRM3 specifically required for binding to known Spen binding partner *SRA* RNA in vitro (*Arieti et al., 2014*). We expressed tagged Spen RRM2-4 in both *Spen* KO mESCs, which express *Xist* upon doxycycline treatment, and in WT male V6.5 mESCs, which do not express *Xist*, but do express *ERVs*. Following UV crosslinking of RNA and protein complexes in vivo, we isolated RRM2-4-bound RNAs using an anti-FLAG antibody and prepared libraries from the isolated RNA.

On mRNAs and lncRNAs, we detected Spen RRM2-4 binding at 29,625 sites (irCLIP-seq clusters), on a total of 4732 transcripts (*Figure 3b,c*; *Supplementary file 1*). Compared to other RBPs, Spen binds relatively very few RNAs in irCLIP-seq (*Figure 3d*). *Xist* was the top bound gene, with much higher binding strength than other RNAs, consistent with its known interaction with the A-repeat region of *Xist* and role in XCI (*Chu et al., 2015*; *McHugh et al., 2015*; *Minajigi et al., 2015*; *Monfort et al., 2015*; *Moindrot et al., 2015*; *Dossin et al., 2020*; *Nesterova et al., 2019*; *Figure 3d*). Spen RRM2-4 binds specifically to the A-repeat monomers as reported for full-length and truncated Spen (*Figure 3e*; *Figure 3—figure supplement 1c*; *Lu et al., 2016*; *Lu et al., 2019*; *Arieti et al., 2014*). This confirmed that our FLAG-tagged RRMs bind Spen's RNA targets in vivo. In addition to *Xist*, Spen binds to several mRNAs, one of which is the *Spen* mRNA itself (*Figure 3d,f*). This is consistent with our observation that loss of Spen protein leads to upregulation of *Spen* mRNA expression, suggesting that the Spen protein represses its own RNA output in *cis*.

To assess whether Spen RRM2-4 binds directly to ERVK-derived RNA, we mapped irCLIP-seq data to TEs in the genome. Because irCLIP-seq reads are short and single ended and thus less likely

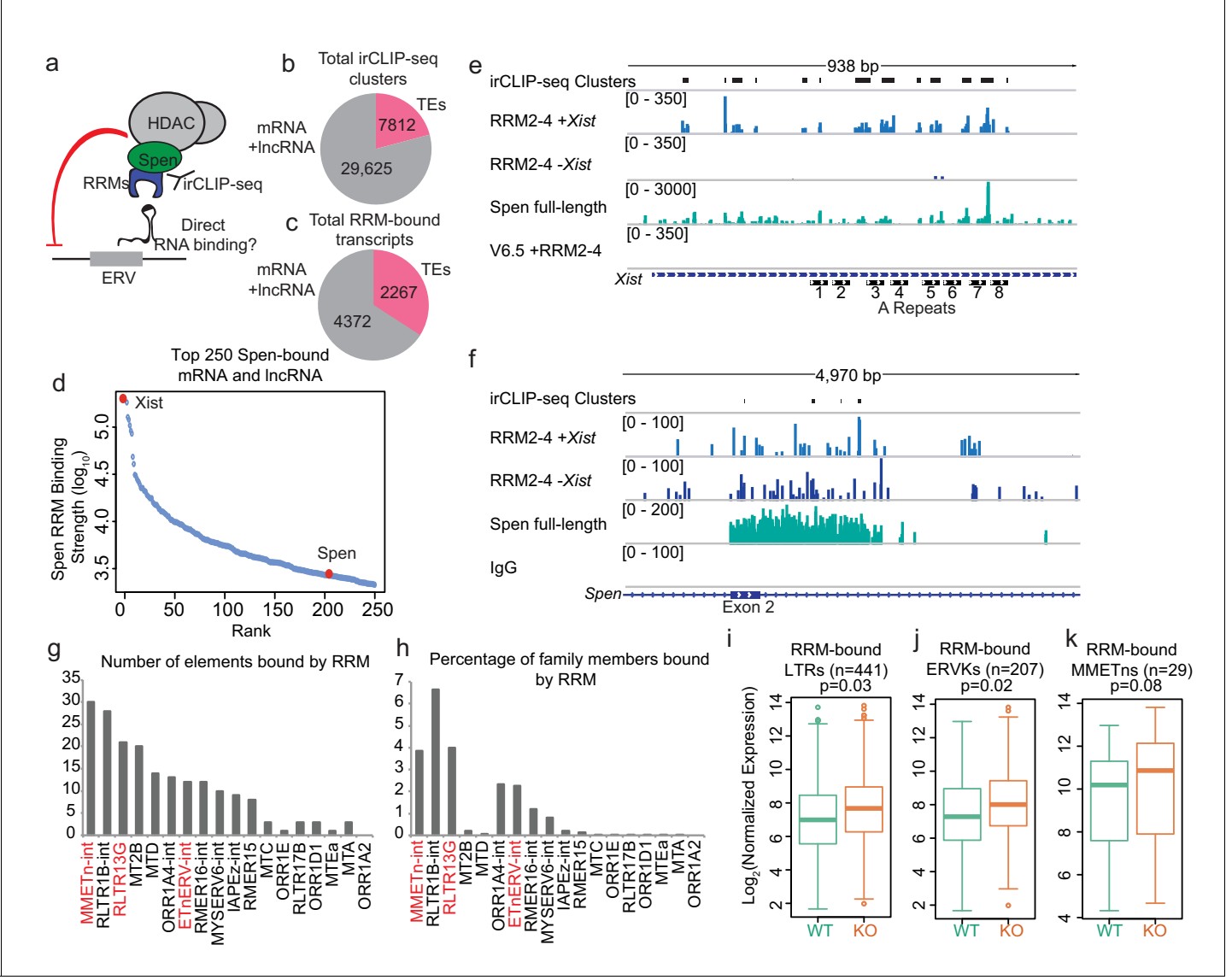

**Figure 3.** Spen's RNA binding domains bind specifically to *ERV* RNAs in vivo. (a) Diagram of irCLIP-seq rationale. irCLIP-seq was performed for the Spen RNA binding domains to identify direct RNA binding partners of Spen genome-wide. (b) Pie chart showing the number of irCLIP-seq clusteres called on TE RNAs and mRNAs and lncRNAs. (c) Pie chart showing the number of RNA transcripts containing at least one irCLIP-seq cluster in TEs and mRNAs and lncRNAs. (d) Binding strength of RRM2-4 for the top 50 Spen-bound RNAs. *Xist* (#1) and *Spen* (#204) are highlighted in red. (e) Tracks showing the irCLIP-seq signal across the A-repeats of *Xist*. Tracks are shown for Spen KO +RRM2-4 (+ / - *Xist*), Spen full-length, and V6.5 male mESCs. irCLIP-seq clusters are shown in black at the top. (f) Tracks showing the irCLIP-seq signal across exon 2 of *Spen*. Tracks are shown for Spen KO +RRM2-4 (+ / - *Xist*), Spen full-length, and an IgG control. irCLIP-seq clusters are shown in black at the top. (g) Graph showing the number of bound TE loci for each TE subfamily with significant Spen RRM2-4 binding. (h) Graph showing the percentage of TE loci bound for each TE subfamily with significant Spen RRM2-4 binding. (i) Expression of RRM-bound LTRs in WT mESCs (green) and *Spen* KO mESCs (orange). Shown is the $\log_2$ transformed normalized RNA-seq read counts. For all boxplots, the thick line represents the median, while the box gives the IQR. *P* values are calculated using a two-tailed T test. (j) Same as in i for RRM-bound ERVKs. (k) Same as in i for RRM-bound MMETns.

The online version of this article includes the following figure supplement(s) for figure 3:

**Figure supplement 1.** irCLIP-seq of RRM2-4 at the *Xist* A-repeats.

**Figure supplement 2.** Comparison of RRM2-4 and Spen full-length binding to ERVs.

**Figure supplement 3.** Motif analysis for RRM2-4 RNA binding sites.

to map uniquely to these repetitive elements, we mapped reads only to the elements that are expressed in our RNA-seq data. We detected a total of 7812 irCLIP-seq clusters on TE RNAs (20% of total clusters detected), covering 2267 individual TEs (*Figure 3b,c*). Of these, we detected RRM2-4 binding at 442 LTRs and 208 ERVKs. The TE subfamily with the greatest number of bound elements is the MMETn-int family, a member of the ETn class (*Figure 3g,h*). In addition, RLTR13G and ETnERV-int showed a large number of elements bound as well as a high percentage of total genomic elements bound (*Figure 3g,h*). RRM-bound elements within these families show modestly increased RNA expression in Spen KO mESCs, consistent with Spen binding contributing to repression of these sites (*Figure 3i–k*). These *ERV* RNAs are bound by RRM2-4 as well as full-length Spen (*Figure 3—figure supplement 1d–f*; *Figure 3—figure supplement 2a,b*).

The specificity of RBP binding to its RNA targets may have a structural and/or sequence basis. In order to understand whether Spen binds to the A-Repeat and *ERV* RNAs in the same manner, we first investigated the sequence features in Spen-bound mRNAs, lncRNAs, and ERVs. We found that RRM2-4 binding sites on mRNA and lncRNAs, as well as ERVs, are enriched for AU content, but did not find any longer, more complex motifs shared amongst the classes. Furthermore, we did not find sequences that showed strong homology to the highly conserved A-Repeat regions (*Figure 3—figure supplement 3a–c*). This was perhaps not surprising, as *Arieti et al., 2014* previously showed that Spen's recognition of the *SRA* lncRNA is largely dictated by RNA secondary structure, not sequence, with Spen binding to a flexible, single stranded region adjacent to a double stranded region.

We asked whether Spen RRMs bind to a similar structural feature in the A-Repeat and in *ERV* RNAs. Spen has previously been reported to bind to the junction between a single stranded loop and a duplex formed by multiple A-Repeat monomers interacting in three dimensions (*Lu et al., 2016*), a structure that is similar to Spen's binding site within SRA (*Arieti et al., 2014*). Upon closer examination of icSHAPE data, which determines whether RNAs are single (high icSHAPE reactivity score) or double stranded (low icSHAPE reactivity score) in vivo (*Spitale et al., 2015*), we found that the A Repeat region of *Xist* consists of a hairpin that contains a small bulging single stranded region, flanked by two larger single stranded loops (*Figure 4a–e*). Both the Spen RRMs and the full length Spen bind to the conserved single stranded regions directly adjacent to the double-stranded hairpin and to the small bulge within the hairpin (*Figure 4a–e*).

To test whether Spen RRMs bind a similar structural motif in *ERVK* RNAs, we analyzed icSHAPE data within repeat regions of the genome (*Sun et al., 2019*). For LTR families for which we had enough loci that are both bound by RRMs and have icSHAPE signal, (RLTR13G, ETnERV, MMETn) we found that Spen binding occurs within short single stranded regions adjacent to double stranded regions (*Figure 4f–h*). This is consistent with a model where Spen RRMs bind to single-stranded bulges within larger duplexes in *ERVK* RNAs (*Figure 4i*; *Figure 4—figure supplement 1a,b*). Taken together, this demonstrates that Spen RRMs bind to the A-repeat of *Xist* and to *ERVK* RNAs in the same manner, recognizing a specific structural feature in the RNA.

Though irCLIP-seq provides a transcriptome-wide map of Spen RRM-RNA binding, we wanted to understand in even more depth the specificity and strength of the interaction between the Spen RRMs and *ERVK* RNAs. We hypothesized that if Spen binds *ERVK*s and the A-Repeat with the same structural recognition mechanism, it should bind with similar affinity and in a competitive manner. We used fluorescence anisotropy experiments to measure the binding affinity of Spen RRMs 2–4 to *ERV* RNA elements as well as the *Xist* A-repeat and the *SRA* RNA H12H13 hairpin, two known interactors of Spen (*Chu et al., 2015*; *McHugh et al., 2015*; *Lu et al., 2016*; *Arieti et al., 2014*). We also measured binding of a version of the Spen RRMs that has five point mutations in RRM3 (RRM3 Mt), which was previously shown to diminish binding of Spen to SRA (*Figure 4—figure supplement 2a*; *Arieti et al., 2014*). Using in vitro irCLIP RNA isolation, we found that the RRM3 Mt binds ~50% less RNA than RRM2-4 confirming these residues are essential for Spen RNA binding (*Figure 4—figure supplement 2b–f*).

Using fluorescence anisotropy, we found that the binding affinity of Spen RRM2-4 for our *ERV* transcripts and the *Xist* A-Repeat are similar (dissociation constant $K_D$ = 53 nM~140 nM), and higher than the affinity of Spen RRM2-4 for the *SRA* hairpin ($K_D$ = 450 nM) (*Figure 5a,b*). The binding affinity of the RRM3 Mt is about half that of the RRM2-4 for both *Xist* A-repeat and the *ERV* RNAs, suggesting that the same residues of Spen's RRM3 domain are binding both *Xist* A-repeat and *ERV* RNAs (*Figure 5b*; *Figure 4—figure supplement 2g*). Finally, we found that an unlabeled *ERV* RNA

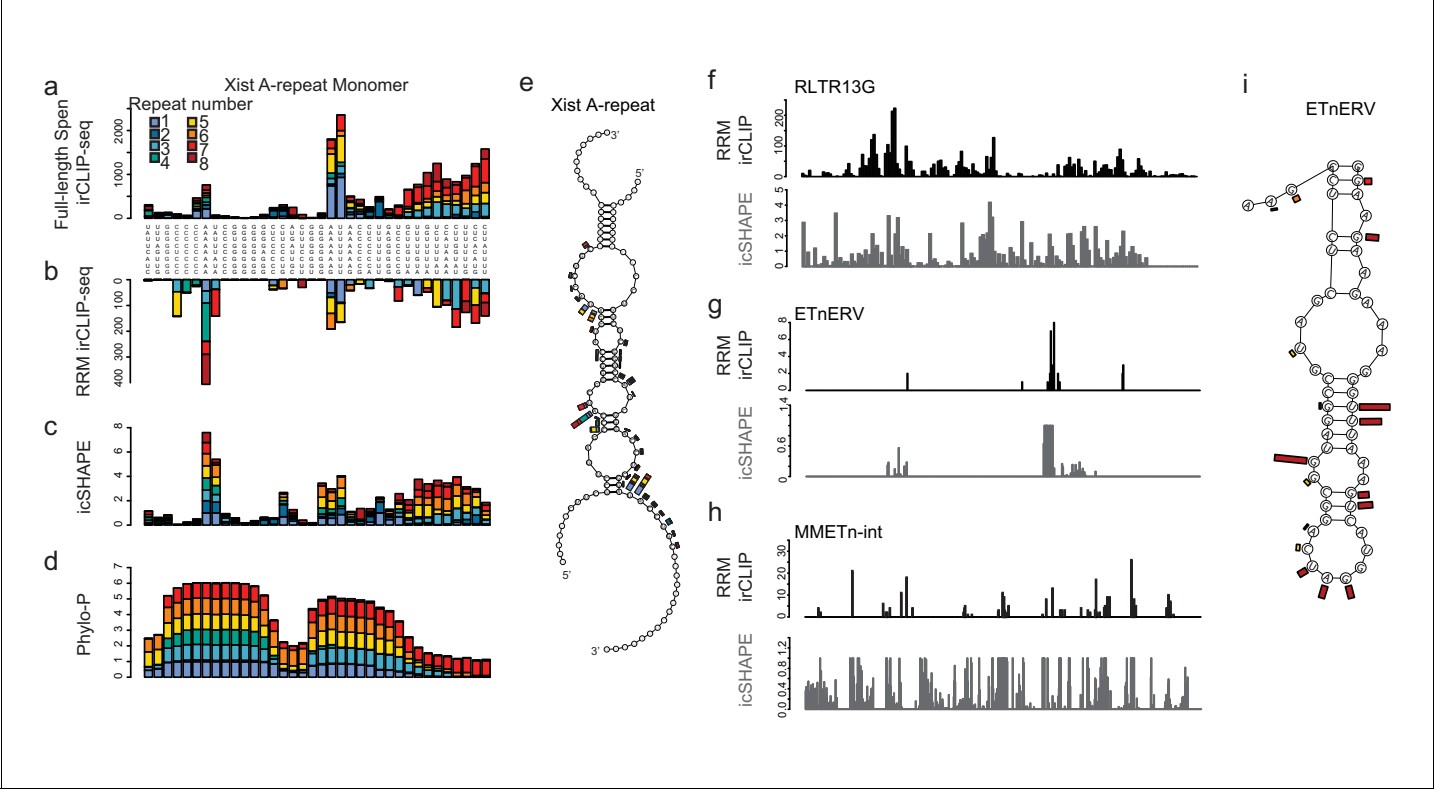

**Figure 4.** Spen RRMs recognize a common structural feature in A-Repeat and *ERV* RNA. (a) irCLIP-seq signal from full-length Spen (*Kahles et al., 2016*) across the 8 *Xist* A-repeats. Signal for each repeat is stacked across the conserved repeat region. (b) Same as in a for RRM2-4 irCLIP-seq. (c) icSHAPE reactivity scores for each of the *Xist* A-repeats performed in vitro. Signal for each repeat is stacked across the conserved repeat region. (d) Phylo-P scores across the A-repeats, showing conservation of repeat sequence at the RRM2-4 binding sites. (e) Structural model for Xist A-repeats 4 and 5 which form a hairpin structure. Structural prediction is supported by icSHAPE data. At each nucleotide, the RRM2-4 irCLIP-seq signal strength is plotted as a bar for each repeat (in color). (f) irCLIP-seq signal (top, black) and icSHAPE reactivity scores (bottom, gray) for the consensus sequence of RLTR13G elements. Included are elements for which we have both confident RRM2-4 binding and icSHAPE data. (g) Same as in f for ETnERV. (h) Same as in f for MMETn-int. (i) Structural model for ETnERV. Structural prediction is supported by icSHAPE data. At each nucleotide, the RRM2-4 irCLIP-seq signal strength is plotted as a bar.

The online version of this article includes the following figure supplement(s) for figure 4:

**Figure supplement 1.** Structural models for Spen-bound *ERV* RNAs.

**Figure supplement 2.** Characterization of RRM3 Mt protein.

competed with a labeled *Xist* A-repeat for binding to Spen RRM2-4, further demonstrating that Spen recognizes these two RNAs, in part, through the same binding domain and in a very similar manner (*Figure 5c*). These data, taken with our irCLIP-seq data, show that Spen binds to ERVK-derived RNA in the nucleus and does so via the same binding mechanism and with the same strength as it binds the A-Repeat of *Xist*. This supports a model in which Spen directly represses ERVK loci in the genome by recognizing their RNA transcripts through their secondary structures.

Collectively, these results show that Spen's RNA binding domains bind TE-derived RNA in mESCs and do so via the same mechanism that they bind *Xist* RNA, with a significant contribution from the RRM3 domain. The observation that the A-repeat may be derived from an ancient ERV insertion (*Elisaphenko et al., 2008*), which shows structural similarity to the ERV elements bound by Spen, suggests that *Xist* evolved its ability to recruit Spen via Spen's recognition of ERV sequence and subsequent recruitment of repressive complexes. Spen itself has evolved relatively little throughout evolution from fish to humans, and Spen's RNA binding domains show no evidence of branch-specific evolution in the eutherian clade (*Figure 6—figure supplement 1*). Thus a mechanism in which the *Xist* RNA itself evolved toward Spen's functionality is fitting.

We reasoned that if the *Xist* A-repeat region evolved via the insertion of a TE that had affinity for Spen into the proto-*Xist* locus, then the insertion of an ERV element into *Xist* may be able to

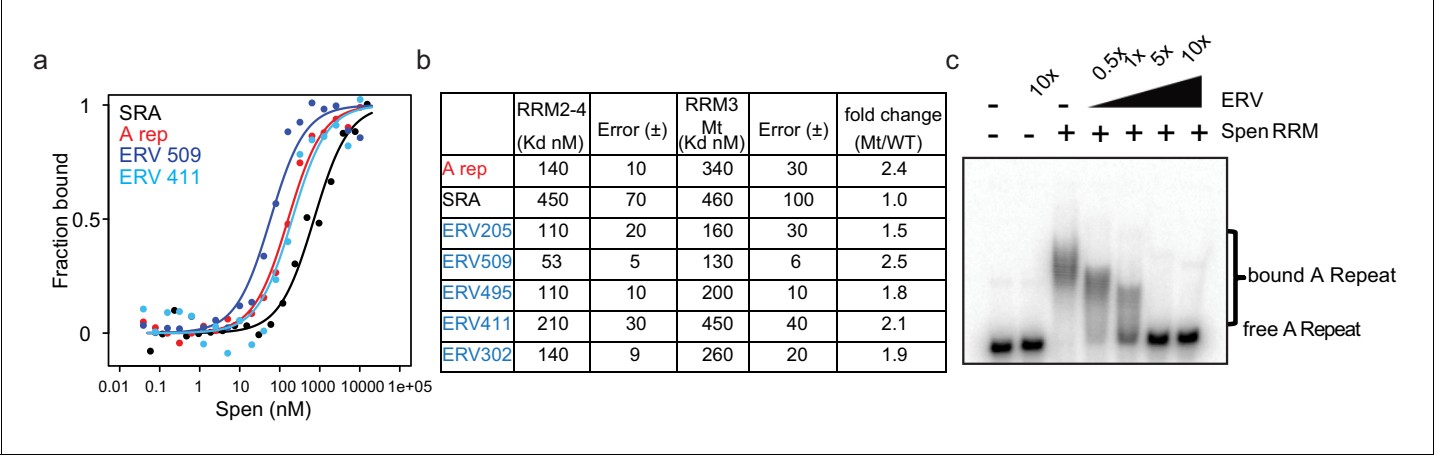

**Figure 5.** ERVs bind Spen RRMs in vitro and compete with the A-Repeat. (**a**) Representative binding curves for Spen RRM2-4 from fluorescence anisotropy experiments. Fluorescence anisotropy values are normalized with saturation and offset parameters from the fit to yield fraction bound. Shown are binding curves for *SRA*, the A-Repeat region of *Xist* (A rep), and two *ERV* RNAs bound in irCLIP-seq data (ERV509, ERV411). (**b**) Direct binding constants ($K_D$) between RNAs of interest and the Spen RRM2-4 and RRM3 Mt proteins from fluorescence anisotropy experiments. Error is the standard error. The fold change in binding between RRM2-4 and RRM3 Mt for each RNA is shown at the right. (**c**) Acrylamide gel image showing competition of ERV509 RNA with $^{32}$P-labeled A-Repeat for binding to Spen RRM2-4. Increasing amounts of unlabeled ERV509 competitor RNA (0.5x, 1x, 5x, and 10x the amount of labeled A-Repeat) was added while the protein and A-Repeat concentrations were kept constant. The higher band depicts Spen-bound A-Repeat, while the lower band shows the unbound A-repeat RNA.

complement the A-repeat deletion of *Xist*. We would expect that this chimeric *Xist-ERV* would be able to 1) recruit and bind to Spen and 2) silence X-linked genes in cis (*Figure 6a*). To test this hypothesis, we utilized a male (XY) mESC line that harbors both a doxycycline inducible *Xist* gene on the X as well as a deletion of the A-repeat region (Xist-ΔA) (*Wutz et al., 2002*). While induction of *Xist* by doxycycline treatment in the WT male mESCs (Xist-WT) leads to silencing of the single X chromosome, *Xist* expression in Xist-ΔA mESCs has no effect on X chromosome gene expression or accessibility (*Wutz et al., 2002*; *Giorgetti et al., 2016*). We used CRISPR-Cas9 genome editing to insert a 9x array of a 154 bp ERV-derived Spen binding site from chromosome 16 into the X chromosome where the A-repeat has been deleted in Xist-ΔA mESCs (Xist-ERV) (*Figure 6—figure supplement 2a–c*). We chose a 9x array of binding sites because the A-repeat region of *Xist* consists of 8.5 repeats in human and 7.5 repeats in mouse, which fold into a complex secondary structure with multiple Spen binding sites (*Lu et al., 2016*; *Minks et al., 2013*). Furthermore, the addition of monomers of the A-repeat up to nine repeats in an A-repeat knockout background leads to a linear increase in chromosome silencing capacity (*Wutz et al., 2002*; *Minks et al., 2013*).

We first asked whether *Xist-ERV* is able to bind directly to and recruit Spen. To do this, we expressed the FLAG-tagged Spen RRM2-4 in our cell lines containing Xist-WT, Xist-ΔA, or Xist-ERV transgenes (two independent clones), and measured *Xist*-RRM binding under optimized expression conditions (more below), by RIP-qPCR. We found that *Xist-ERV* is bound by Spen RRM2-4, similar to *Xist-WT* and in contrast to *Xist-ΔA* which could not bind Spen (*Figure 6b*). Next, when we quantified the ability to induce *Xist-ERV* expression with the addition of doxycycline, we were surprised to find that expression of *Xist-ERV* could not be induced, in comparison to *Xist-WT* and *Xist-ΔA* (*Figure 6c*). Genetic analysis confirmed that the Tet operator and CMV promoter were intact in all of our *Xist-ERV* clones (*Figure 6—figure supplement 2d*), suggesting that the lack of gene induction may be due to epigenetic silencing recruited by Spen. To test whether the inability to induce *Xist-ERV* expression was due to silencing in cis, we treated cells with an HDAC1/2 inhibitor (HDAC1/2i) or an HDAC3 inhibitor (HDAC3i) for 24 hr before *Xist* induction. We found that both HDAC1/2i and HDAC3i de-repressed the transgene and *Xist-ERV* was expressed (*Figure 6c*; *Figure 6—figure supplement 2e–h*).

To test whether this *Xist-ERV* silencing was Spen-dependent, we used siRNAs to knockdown Spen 24 hr prior to induction of *Xist* with doxycycline. We found that *Xist-ERV* expression was elevated in Xist-ERV clones 1 and 2 after Spen knockdown (*Figure 6d*). The rescue of *Xist-ERV*

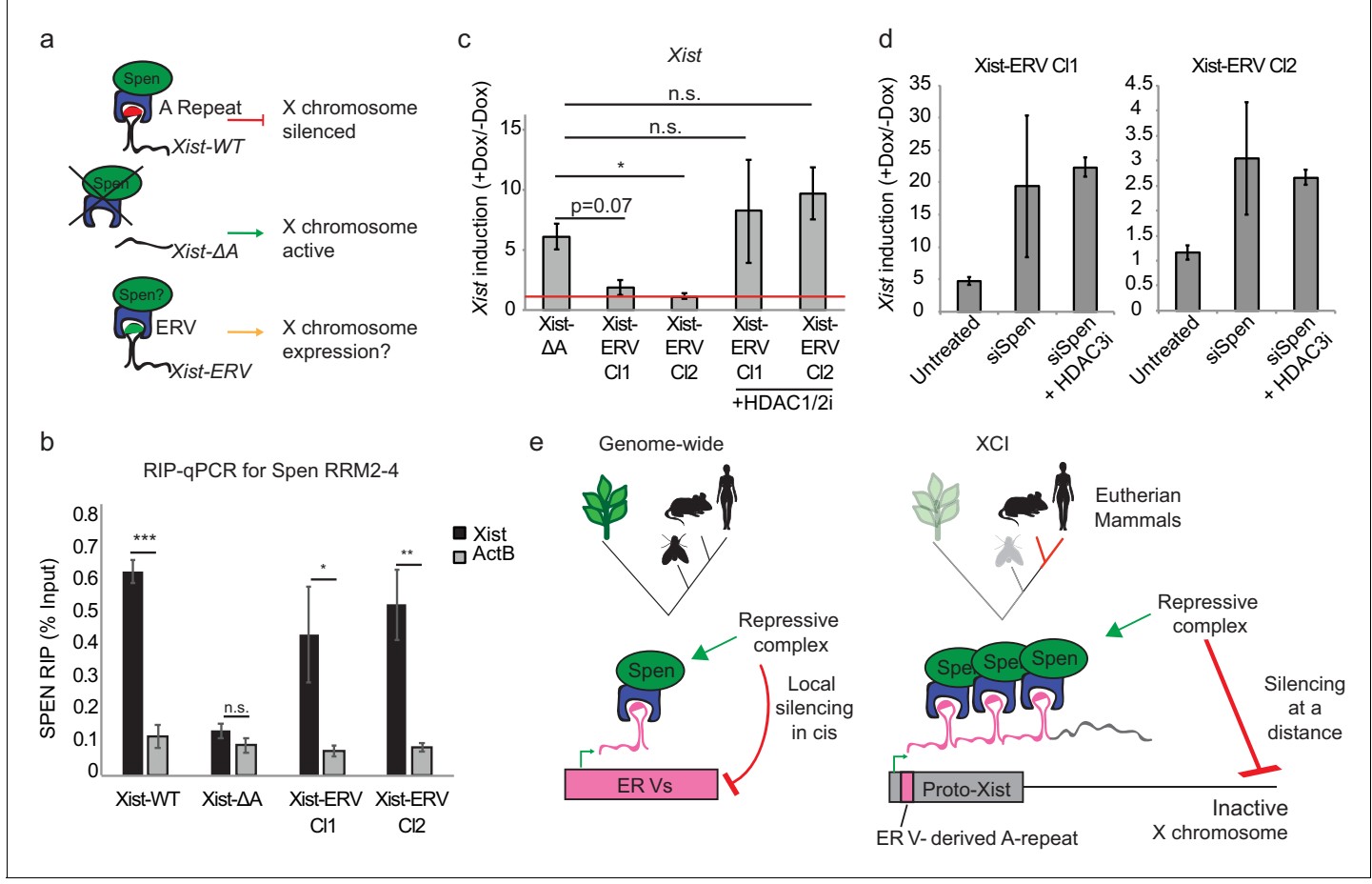

**Figure 6.** Insertion of an ERV element in place of A-repeat restores Spen binding and local Xist silencing. (**a**) Diagram showing the experimental rationale for replacing the A-Repeat with a 9x tandem insert of an ERV-derived Spen binding site. (**b**) *Xist-ERV* restores Spen recruitment. RIP-qPCR for *Xist* and *ActB* in Xist-WT, Xist-ΔA, and Xist-ERV clones 1 and 2 mESCs. qRT-PCR values are normalized to input samples. Error bars represent standard error for four independent biological replicates. p values are derived from two-tailed *t*-test. (**c**) Xist-ERV causes epigenetic silencing of the *Xist* locus. *Xist* expression level in +Dox compared to –Dox conditions for Xist-ΔA mESCs and Xist-ERV mESCs with and without HDAC1/2 inhibitor treatment. *Xist* levels are first normalized to *ActB* and then to –Dox levels. The red line is drawn at y = 0, representing no increase in expression of *Xist* with the addition of doxycycline. For each condition, four independent experiments were conducted. p values are derived from two-tailed T test. (**d**) Xist-ERV silencing requires Spen and HDAC3. *Xist* expression level in +Dox compared to –Dox conditions for Xist-ΔA mESCs and Xist-ERV mESCs with and without siRNA-mediated knockdown of Spen or siRNA-mediated knockdown of Spen + HDAC3 inhibition. For each condition, four independent experiments were conducted. (**e**) Model for how Spen evolved to bind *Xist* through its *ERV* binding mechanism in eutherian mammals. The online version of this article includes the following figure supplement(s) for figure 6:

**Figure supplement 1.** Spen's RRM domains have not evolved specifically in the Eutherian lineage.
**Figure supplement 2.** Insertion of an ERV element in the A-Repeat locus.
**Figure supplement 3.** Characterization of *Xist-ERV* function.

expression by Spen depletion was quite variable from experiment to experiment; addition of an inhibitor of HDAC3, which is recruited by Spen (*Żylicz et al., 2019*; *McHugh et al., 2015*), made the Xist-ERV induction substantially more consistent and up to 20-fold in at least one clone (*Figure 6d*). This suggested that *Xist-ERV* recruits Spen, leading to local silencing of the *Xist* transgene in cis.

Therefore, the insertion of an ERV-derived RNA motif into *Xist* is sufficient to partially substitute for the A-repeat, recruit Spen to the chimeric RNA, and mediate strictly local gene silencing in cis. This is the first demonstration of how a TE insertion during evolution could confer new functionality on a noncoding RNA, by introducing a new protein recruitment domain (*Johnson and Guigó, 2014*).

Finally, we tested whether this Spen recruitment by the *Xist-ERV* RNA is sufficient to lead to chromosome-wide silencing. We conducted these experiments in the presence of HDAC1/2 inhibition, which increases *Xist* induction but was previously shown not to affect silencing during XCI (*Żylicz et al., 2019*). While expression of X-linked genes *Pgk1*, *Gpc4*, *Mecp2*, and *Rnf12* was reduced after *Xist* induction in Xist-WT cells, expression was unchanged upon expression of *Xist-ΔA* or *Xist-ERV* (*Figure 6—figure supplement 3a*). Thus, while the addition of the ETn array to an A-repeat-deficient *Xist* was sufficient to recruit Spen, it was not sufficient to carry out XCI. These experiments were conducted in the presence of HDAC1/2i, necessary to overcome Xist-ERV silencing of its own locus, but may also inhibit the chimeric RNA's silencing of distal loci. Furthermore, this could be due to the differences in levels of *Xist* that we are able to induce in this system, which are lower in the Xist-ERV cells than the Xist WT or Xist ΔA cells, even in the presence of HDAC1/2i. Nevertheless, our results are consistent with the hypothesis that insertion of an ERV element into the proto-*Xist* locus was an *early* evolutionary step in the functionalization of the A-repeat, but that additional rounds of evolution may have subsequently acted on this region of the genome in order to evolve *Xist*'s full functionality for silencing the entire chromosome over long genomic distance (*Elisaphenko et al., 2008*; *Brockdorff, 2018*).

## Discussion

Here we show that Spen, a highly conserved and pleiotropic RNA binding protein, binds to and regulates specific classes of ERVK loci in mouse embryonic stem cells. Spen represses these ERVK loci via a direct mechanism where it binds transcripts derived from these parasitic elements, and can then recruit its chromatin silencing protein partners. In the absence of Spen, these elements become derepressed, showing loss of repressive H3K9me3 and gain of active chromatin modifications. Sensing transposon silencing at the RNA level is attractive, as it provides a mechanism for the cell to identify the 'leaky' locus and target the silencing machinery to the locus in *cis*.

These observations are striking when taken with the observation that the *Xist* RNA, which is also robustly bound by Spen, is derived from a number of ancient retroviral insertions (*Elisaphenko et al., 2008*). We show direct evidence, for the first time, that an *Xist* binding protein can bind to ERV-derived RNA in vivo, and thus may have been recruited to *Xist* via the insertion of an ERV element into the proto-*Xist* locus (*Figure 6e*). This evidence supports the hypothesis that lncRNAs, which are evolutionarily young, may exapt TE-protein interactions in order to form functional protein binding domains (*Johnson and Guigó, 2014*; *Kelley and Rinn, 2012*; *Guttman et al., 2009*; *Hezroni et al., 2015*). In effect, *Xist* lncRNA 'tricks' the female cell to perceive the inactive X chromosome as being coated by dangerous transposon RNAs. The cell can then deploy Spen, a transposon-silencing protein, to carry out X chromosome dosage compensation.

One of the major mysteries of X inactivation is the *cis*-restriction of *Xist*, which only silences the chromosome from which it is expressed, a property critical for dosage compensation of two X chromosomes in females to one X in males. Our discovery of the link between Spen and transposon silencing suggests an ancient evolutionary origin for *cis* silencing. Spen sensing of *ERV* RNA and Spen-mediated silencing are *cis* restricted, and indeed grafting ERV into the *Xist* locus caused *cis* silencing of *Xist* RNA expression itself. Hence, these data suggest that the invasion of an ERV-like sequence into proto-*Xist* led first to RNA-based silencing in *cis*. Subsequent mutations in this proto-*Xist* may then have allowed this RNA to localize to and silence distal sites on the X chromosome. The transition from short to long range silencing in *cis* may be a fruitful topic in future studies.

## Materials and methods

**Key resources table**

| Reagent type (species) or resource | Designation | Source or reference | Identifiers | Additional information |
|---|---|---|---|---|
| Gene (mouse) | Spen | RefSeq | NM_019763.2 | |

*Continued on next page*

*Continued*

| Reagent type (species) or resource | Designation | Source or reference | Identifiers | Additional information |
|---|---|---|---|---|
| Cell line (mouse) | TXY WT and TXY deltaA | Dr. Edith Heard | Xist-WT and Xist-deltaA | Cell lines have a doxycycline inducible Xist gene + / - the A repeat region |
| Cell line (mouse) | HATX WT and HATX Spen KO | Dr. Anton Wutz | WT, *Spen* KO | KO cells have a deletion of the majority of the Spen gene. They also have a doxycycline inducible Xist transgene on the X chromosome |
| Antibody | Goat polyclonal Anti-Oct4 | Santa Cruz | Sc-8629 | 7.5 ug/ChIP |
| Antibody | Rabbit polyclonal Anti-H3K27Ac | Active Motif | 39133 | 5 ug/ChIP |
| Antibody | Rabbit polyclonal Anti-SHARP(Spen) | Bethyl | A301-119A | Antibody to endogenous Spen |
| Antibody | Mouse monoclonal Anti-FLAG M2 | Sigma | F3165 | Antibody against FLAG tag on Spen RRM2-4 |
| Antibody | Rabbit polyclonal Anti-H3K9me3 | abcam | AB8898 | 5 ug/ChIP |
| Antibody | Rabbit polyclonal Anti-H3K4me3 | Active Motif | 39159 | 5 ug/ChIP |
| Sequence-baed reagent | mXist 570 RNA FISH probe set | Stellaris | SMF-3011–1 | 125 nM concentration |
| Commercial assay or kit | Nextera DNA prep kit | Illumina | FC-131–1002 | Used for ATAC-seq and for ChIP-seq library preparation. |
| Chemical compound, drug | HDAC1/2 inhibitor, BRD6688 | Cayman Chemical | 1404562-17-9 | Inhibitor of HDAC1 and 2. Used at 10 uM. |
| Chemical compound, drug | HDAC3 inhibitor, RGFP966 | Sigma | SML1652 | Inhibitor of HDAC3. Used at 10 uM. |
| Sequence-based reagent | Xist Primers | Dr. Edith Heard | 5'GCTGGTTCGTCTATCTTGTGGG3' 5'CAGAGTAGCGAGGACTTGAAGAG3' | |

## Cell lines and culture

HATX WT and Spen KO clone 2 and clone 3 cell lines were gift of A. Wutz and are described in detail in *Monfort et al., 2015*. TXY WT (Xist-WT) and TXY deltaA (Xist-ΔA) cell lines were gift of E. Heard. mESCs were cultured in serum/LIF conditions (Knockout DMEM (GIBCO 10829018), 1% Penicillin/streptomycin, 1% GlutaMax, 1%non-essential amino acids, beta-mercaptoethanol, Hyclone fetal bovine serum (Fisher SH3007103), and 0.01% LIF (ESGRO ESG1107). Cells were cultured on 0.2% gelatin-covered plates and passaged every 2–3 days with trypsin as needed. All cell lines tested negative for mycoplasma.

## *Xist* induction

For *Xist* induction experiments, mESCs were plated at a density of 75,000 cells/well in a 6-well plate. 24 hr after plating, cells were treated with 2 ug/mL doxycycline for 48 hr (unless shorter or longer time is indicated in figure legend). Induction of *Xist* was confirmed using qPCR and *Xist* RNA FISH.

qPCR primers used for Xist have the following sequences: Forward 5'GCTGGTTCGTCTATCTTG TGGG3' and Reverse 5'CAGAGTAGCGAGGACTTGAAGAG3'.

## HDAC inhibition experiments

For HDAC inhibition experiments, cells were treated with either HDAC1/2i (BRD6688, Cayman Chemical Company, 1404562-17-9) or HDAC3i (RGFP966, Sigma SML1652) at 10 uM final concentration. HDAC inhibition was initiated 24 hr prior to induction of Xist with doxycycline and continued for 24 hr during Xist induction (48 hr total). –HDACi samples were treated with DMSO as a control.

## *Xist* RNA FISH

Following Xist induction, adherent cells on coverslips were fixed using 4% paraformaldehyde for 10 min. Cells were permeablized in 0.5% triton for 10 min on ice and then stored in 70% EtOH. After dehydration in EtOH, cells were rehydrated in wash buffer (2xSSC + 10% formamide) for 5 min. Probe (Stellaris mXist 570, SMF-3011–1) was then added at a concentration of 125 nM (in wash buffer + 10% Dextran Sulfate) and incubated overnight at 37'C. Following washes in wash buffer + formamide, coverslips were mounted on slides with Vectashield + DAPI.

## NPC differentiation

NPC differentiation from mESCs was performed as previously described (*Conti et al., 2005*). Briefly, mESCs were plated on gelatin-coated plates in N2B27 medium for 7 days. On day 7, cells were dissociated with Accutase and cultured in suspension in N2B27 medium with FGF and EGF (10 ng/ml, each). On day 10, embryoid bodies were plated onto 0.2% gelatin-coated plates and allowed to grow until 14 days. At 14 days, cells were stained with anti-Nestin (Millipore, MAB353) antibody to confirm NPC identity. For qRT-PCR RNA was extracted from Trizol using the RNEasy Mini kit (Qiagen). RNA was then used directly for qRT-PCR using the following primers. Oct4-F: 5' TTGGGC TAGAGAAGGATGTGGTT3', Oct4-R: 5' GGAAAAGGGACTGAGTAGAGTGTGG3' ActB-F: 5'TCC TAGCACCATGAAGATCAAGATC3', ActB-R: 5'CTGCTTGCTGATCCACATCTG3'.

## Immunofluorescence

Immunofluorescence for Oct4 and Nestin was performed on adherent cells fixed with 4% paraformaldehyde. After fixation, cells were permeablized with 0.5% PBS/Triton for 10 min on ice. Following permeablization, cells were blocked in 10% FBS in 0.1% PBS/Tween for 1 hr at room temperature. After blocking, cells were incubated for 2 hr in primary antibody (Nestin: Millipore, MAB353, Oct4: Santa Cruz sc-8629), washed in 0.1% PBS/Tween and then incubated for 2 hr in secondary antibody (Goat anti-mouse 488 Life Technologies ab150113, Rabbit anti-goat 488 Life Technologies A27012). After washing, cells were stained with DAPI and imaged at 40x magnification.

## Chromatin immunoprecipitation

Cells were fixed with 1% formaldehyde for 10 min at room temperature and subsequently quenched with 0.125M glycine. Cells were then snap frozen and stored at −80'C. Cells were then lysed (50 mM HEPES-KOH, 140 mM NaCl, 1 mM EDTA, 10% glycerol, 0.5% NP-40, 0.25% Triton X-100) for 10 min at 4'C. Nuclei were lysed (100 mM Tris pH 8.0, 200 mM NaCl, 1 mM EDTA, 0.5 mM EGTA) for 10 min at room temperature. Chromatin was resuspended in sonication buffer (10 mM Tris pH 8.0, 1 mM EDTA, 0.1% SDS) and sonicated using a Covaris Ultrasonicator to an average length of 220 bp. For all ChIPs, 5 million cells per replicate were incubated with 5 ug (Histone) or 7.5 ug (Oct4) antibody overnight at 4'C (antibodies: H3K9me3: abcam AB8898; H3K27Ac: Active Motif 39133; H3K4me3: Active Motif 39159; Oct4: Santa Cruz sc-8629). Antibody-bound chromatin was incubated with Protein G Dynabeads (Invitrogen, 10004D) for 4 hr at 4'C and eluted in Tris buffer (10 mM Tris pH 8.0, 10 mM EDTA, 1% SDS). Crosslinks were reversed by incubation overnight at 65'C followed by treatment with 0.2 mg/mL proteinase K (Life Technologies, AM2548) and 0.2 mg/mL RNAse A (Qiagen). DNA was purified using Qiagen Minelute Columns (Qiagen, 28006). For library preparation, 4 ng ChIP DNA was incubated with transposase (Illumina, Fc-121–1030) for 10' at 55'C. DNA was then amplified using Nextera barcoding adapters and sequenced on an Illumina Nextseq (2 × 75 bp reads).

## ChIP-seq analysis

ChIP-seq libraries for histone modifications were sequenced on an Illumina NextSeq with paired-end 75 bp reads. Reads were mapped to the mm9 genome build using Bowtie2 (*Langmead and Salzberg, 2012*). Unique and coordinated mapped paired reads were kept and duplicate reads were then removed using Picard Tools.

Peaks were called using MACS2 with a q-value cutoff of 0.01 with input as background (*Zhang et al., 2008*). Peaks called from two technical replicates were compared by IDR package and further filtered by the Irreproducible Discovery Rate(<0.05) (*Li et al., 2011*). Reproducible peaks from WT and KO conditions were merged and reads in these peaks were counted across all libraries. Differential peaks were called using reads counts from reproducible peaks with DESeq2(fold change >= 2 and Padjust <0.01) (*Love et al., 2014*). Bigwig files were generated from deduplicated bam files using bedtools for visualization (*Quinlan and Hall, 2010*).

## ATAC-seq

ATAC-seq library preparation was performed exactly as described in *Buenrostro et al., 2013*. Briefly, ESCs were dissociated using Accutase (SCR005,Millipore). 50,000 cells per replicate (two replicates per clone) were incubated with 0.1% NP-40 to isolate nuclei. Nuclei were then transposed for 30' at 37'C with adapter-loaded Nextera Tn5 (Illumina, Fc-121–1030). Transposed fragments were directly PCR amplified and sequenced on an Illumina NextSeq 500 to generate 2 × 75 bp paired-end reads.

## ATAC-seq analysis

Reads were trimmed using CutAdapt (*Martin, 2011*) and mapped to the mm9 genome build using Bowtie2 (*Langmead and Salzberg, 2012*). Unique and coordinated mapped paired reads were kept and duplicate reads were then removed using Picard Tools. Peaks were called using MACS2 with a q-value cutoff of 0.01 and no shifting model (*Zhang et al., 2008*). Peaks called from two technical replicates were compared by IDR package and further filtered by the Irreproducible Discovery Rate (<0.05) (*Li et al., 2011*). Reproducible peaks from WT and KO conditions were merged and reads in these peaks were counted across all libraries. Differential peaks were called using reads counts from reproducible peaks with DESeq2(fold change >= 2 and P-adjusted <0.01) (*Love et al., 2014*). To calculate $P$ values in DESEq2, for each gene the counts were modelled using a generalized linear model (GLM) of negative binomial distribution among samples. The two-sided Wald statistics test was processed for significance of the GLM coefficients. The Benjamini-Hochberg correction was applied to all p-values to account for multiple tests performed. This gives the final adjusted p-values used for assessing significance.

## ATAC-seq analysis for X chromosome silencing

ATAC-seq reads were trimmed and mapped as described above. Peaks were called in all samples (+ and – doxycycline) and bedtools multicov was used to count the number of reads falling within each peak in each sample. In order to avoid bias in normalization in the case of a global loss of accessibility on the X chromosome, we first normalized count data using all peaks genome-wide. We then calculated the ratio of average normalized reads in +dox / -dox samples at each peak on the X chromosome.

## RNA-seq

RNA-seq library preparation was performed using the TruSeq Stranded mRNA Library Prep Kit (Illumina, RS-122–2102). In place of poly-A selection, we performed Ribosomal RNA depletion using the RiboMinus Eukaryote System v2 (Life Tech. A15026). Each library was prepared from 200 ng starting RNA, and two replicates were made for each cell line. RNA-seq libraries were sequenced on an Illumina HiSeq 4000 with paired 2 × 75 bp reads.

## RNA-seq analysis

RNA-seq reads were mapped to the mm9 genome using STAR (*Dobin, 2016*). Following read alignment, reads were assigned to transcripts using FeatureCounts in R (RSubread) (*Liao et al., 2014*). GTF file from gencode was used for feature assignment (https://www.gencodegenes.org/mouse_

releases/1.html). Differential genes were called using DESeq2 (fold change >= 2, P-adjusted <0.001) (*Love et al., 2014*). For GO term analysis of RNA-seq data, GOrilla was used (*Eden et al., 2009*).

## Enrichment of genomic loci in ATAC-seq and ChIP-seq data

HOMER was used for enrichment of genomic locations in ATAC-seq and ChIP-seq data (*Heinz et al., 2010*). Specifically, we used the annotatePeaks.pl script with the –genomeOntology option.

## In vivo irCLIP-seq

For in vivo irCLIP-seq experiments, V6.5 male mESCs, or *Spen* KO Clone 2 mESCs were transduced with a lentivirus carrying Spen RRMs 2–4 with the SV40 nuclear localization signal, 2x HA tags, and 3x FLAG tag ('SpenRRM-FLAG'; pLVX-EF1a-SV40NLS-SpenRRM234-2xHA-3xFLAG-IRES-zsGreen). Following transduction, GFP+ cells were FACS sorted and single colonies were picked. Western blotting was used to confirm the expression of SpenRRM-FLAG at 37 kDa. irCLIP was performed exactly as described (*Zarnegar et al., 2016*). Briefly, 2 million mESCs were UV-crosslinked (254 nM UV-C) at 0.3 J/cm$^2$, and then lysed (1% SDS, 50 mM Tris pH7.5, 500 mM NaCl) and sonicated using a Bioruptor (Diagenode) (six cycles 30' on 45' off). Clarified lysates were incubated overnight with Protein A dynabeads conjugated to mouse anti-FLAG (Sigma Aldrich F3165) or anti-IgG antibody overnight at 4C. Following IP, beads were washed in high stringency buffer (15 mM Tris-HCl, pH 7.5; 5 mM EDTA; 1% Triton X-100; 1% Na-deoxycholate; 0.001% SDS; 120 mM NaCl; 25 mM KCl), high salt buffer (15 mM Tris-HCl, pH 7.5; 5 mM EDTA; 1 mM Triton X-100; 1% Na-deoxycholate; 0.001% SDS; 1000 mM NaCl), and low salt buffer (15 mM Trish-HCl, pH 7.5; 5 mM EDTA). After washing, RNA was digested using RNase I at 1:2000. RNA ends were then dephosphorylated and IR-conjugated linker (/5Phos/AGATCGGAAGAGCGGTTCAGAAAAAAAAAAAA/iAzideN/AAAAAAAAAAAA/3Bio/) was ligated overnight. Following ligation, samples were run on a 4–12% Bis-Tris gel, transferred to nitrocellulose and imaged using an Odyssey LiCOR scanner. RNA-protein complexes were eluted from nitrocellulose and treated with proteinase K to eliminate protein. Following RNA elution, Oligo(dT) Dynabeads were added to capture the polyadenylated irCLIP adapter. Next, the TruAmp sequencing adapter was added and RNA was reverse transcribed using SuperScript IV. Following reverse transcription, DNA was captured with streptavidin beads, circularized on the bead, and then PCR amplified. irCLIP libraries were then purified using PAGE gel purification and sequenced on an Illumina NextSeq using single-end 75 bp reads.

## Expression and purification of wild-type and mutant spen RRMs 2–4 for in vitro irCLIP RNA binding

The following constructs for SPEN (Split Ends protein), SHARP (human homolog, SMRT/HDAC1 Associated Repressor Protein), were designed and cloned into pET28a for *E. coli* expression.

  SPEN-1=6 His-GST-TEV- RRM 2,3,4-SG-Flag
  SPEN-4=6 His-GST-TEV- RRM 2,3,4-RRM3mutant-SG-Flag

DNA for each was transformed into BL21 (DE3) Star competent cells, the transformation mix plated on LB-Kan agar and single colonies obtained the following day. 4 × 50 ml overnight pre-cultures were setup with single colonies for each construct in Terrific Broth (Teknova) + Kanamycin (50 ug/ml) (TB-Kan), at 37C, 250 rpm. Pre-cultures were used to inoculate 1li volumes of TB-Kan expression media at 10 ml/li (gives a starting OD600 = 0.15). Expression cultures were incubated at 37C, shaken at 250 rpm, for 4 hr then chilled to 18C for 1 hr and induced overnight with 0.25 mM IPTG at 18C. Cultures were harvested and resulting cell pellets were stored frozen at −80C. Frozen cell pellets were resuspended in 5 ml/g of Lysis Buffer(50 mM HEPES;pH 7.5, 300 mM NaCl, 20 mM Imidazole, 0.1% Triton X-100, DNase I (1 ug/ml), Lysozyme (1 ug/ml), 5 mM 2-ME, 2 x Roche protease Inhibitor tablet/50 ml and 1 mM PMSF). Resuspended pellets (in suspensions) were homogenized by Polytron and lysed by two rounds through a Microfluidizer (18000 psi). Lysates were clarified by centrifugation at 33000 x g for 60mins, the supernatant extracted and subjected to batch binding with 5 ml of NiNTA (Qiagen) for 2 hr, 4C with end-to-end rotation. Resin was washed with 20Cv's of Lysis Buffer containing 20 mM Imidazole, followed by elution of bound protein with 3 × 1 CV of Elution Buffer (Lysis Buffer + 250 mM Imidazole). Eluted fractions were pooled and dialysed overnight against 4li of 50 mM Hepes;pH 7.5, 300 mM NaCl, 1 mM TCEP with the addition of TEV protease to

effect in-situ cleavage of the N-terminal 6His-GST tag. TEV cleaved solution was passed through a 0.5 ml NiNTA and the flow-through collected and concentrated 3–5 fold to give 4–10 mg/ml samples which were aliquotted and stored frozen at −80C.

## In vitro irCLIP RNA binding

In vitro irCLIP-seq was performed similarly to in vivo irCLIP with the following modifications. First, nuclear RNA was isolated as previously described (*Gagnon et al., 2014*). Briefly, cells were pelleted and resuspended in Hypotonic Lysis Buffer (10 mM Tris pH 7.5, 10 mM NaCl, 3 mM MgCl$_2$, 0.3% NP-40 and 10% glycerol) with 100 Units SUPERase-In. Cells were incubated on ice for 10 min, vortexes, and centrifuged for 3 min at 1000xg at 4C. The pelleted nuclei were then washed 3x in hypotonic lysis buffer and centrifuged for 2 min at 200xg at 4C. Pellet was then resuspended in TRIzol and RNA extraction was carried out per the manufacturer's instructions. 20 μg nuclear RNA was then fragmented (Ambion, AM8740), and incubated with 5 μg recombinant Spen RRMproteins (RRM2-4 WT or RRM2-4:RRM3 Mt) at 4C for 2 hr. RNA-protein complexes were then transferred to a 3 cm cell culture dish on ice and crosslinked (254 nM UV-C) at 0.3 J/cm$^2$. Following crosslinking, RNA-protein complexes were incubated with antibody-bound Protein A Dynabeads (anti-FLAG [Sigma Aldrich F3165] or anti-IgG) at 4C for 2 hr. Following IP, beads were washed with IPP buffer (50 mM Tris-HCl pH 7.4, 100 mM NaCl, 0.05% NP-40, 5 mM EDTA), low salt buffer (10 mM Tris-HCl pH7.4, 50 mM NaCl, 1 mM EDTA, 0.1% NP-40), and high salt buffer (10 mM Tris-HCl pH7.4, 500 mM NaCl, 1 mM EDTA, 0.1% NP-40, 0.1%SDS). After washes, the protocol was performed exactly as in in vivo irCLIP from dephosphorylation to membrane imaging(above).

## irCLIP quality control analysis of endogenous SPEN

irCLIP was performed as previously described (*Zarnegar et al., 2016*). In brief, 50% confluent WT and KO mESCs were UV-C crosslinked on ice with 0.3 J/cm2 and then lysed on-plate with 1 ml of lysis buffer (20 mM Tris, pH 7.5, 100 mM NaCl, 1% Igepal CA-630, 0.5% NaDeoxycholate, 1 mM EDTA). Lysates were sonicated for 10 s (1 s on 1 s off) at 10% power with a Branson Digital Sonifier and clarified for 10 min at 13,500 rpm. Protein concentrations were quantified with the Pierce BCA assay. For RNAse digests, ice cold lysates were adjusted to 2 mg/ml and incubated with RNAse1 at a final concentration of 0.025 u/ml for 7.5 min at 37C. Reactions were stopped by addition of 1 ml of ice cold lysis buffer. For all immunoprecipitations, 3 ug of control IgG or anti-SPEN antibody were incubated overnight with 500 ug of lysates containing 30 ul of Protein G Dynabeads. Following irCLIP washes, immunoprecipitations were 3' dephosphorylated for 30 min and then ligated overnight with 1 pmole of irCLIP adapter. Immunoprecipitations were then rinsed once with ice cold PBS and then resuspended in 1 x LDS buffer containing 10 mM DTT and heated at 75C for 15 min prior to SDS-PAGE.

## Analysis of expressed repeats in RNA-seq data

To identify expressed transposable elements in RNA-seq data, we counted all unique reads mapping to the database of repeat element loci from RepeatMasker for the mm9 genome. Bedtools multicov was used to count the number of reads per sample in each element. Read counts were then converted to fpkm. All elements with an FPKM >= 1 in all samples were considered expressed and used for downstream analysis.

## ERV knock-in at the *Xist* A repeat

For Xist rescue experiments with ETn elements, we utilized the TXY ΔA cells which harbor a doxycycline inducible Xist transgene with the A Repeat region deleted. We designed donor plasmids for homology directed repair containing homology arms to the transgenic Xist present in TXY ΔA cells that contain a 154 bp portion of an ETn that binds to Spen, repeated 9x. Homology arms flanking the ETn insertions were cloned into the PCR4-Blunt TOPO vector (Thermo Fisher K287520). Cells were then nucleofected with a Cas9 RNP complex directed to the A Repeat region. 10 ug Cas9 protein (PNA Bio CP01-20) was mixed with 10 ug modified guideRNA (Synthego, GCGGGATTCGCC TTGATTTG) and 20 ug HDR donor vector. Cells were nucleofected with RNP complex and donor vector using the Amaxa Nucleofector (Lonza VPH-1001). Clonal colonies were picked and genotyped

by PCR followed by Sanger sequencing. PCR of the Tet operator array was done using the following primers: TetO-F: 5'CCTACCTCGACCCGGGTACC3', TetO-R: 5'GGCCACTCCTCTTCTGGTCT3'.

## RIP-qPCR

TXY WT, TXY ∆A, and TXY-ERV Clones 1 and 2 were transduced with a lentivirus carrying Spen RRMs 2–4 with the SV40 nuclear localization signal, 2x HA tags, and 3x FLAG tag ('SpenRRM-FLAG'; pLVX-EF1a-SV40NLS-SpenRRM234-2xHA-3xFLAG-IRES-zsGreen). Following transduction, GFP+ cells were FACS sorted and expression of the RRM-FLAG construct was confirmed by Western Blot using an anti-FLAG antibody (Sigma F3165). SpenRRM-FLAG expressing cells were treated with 10 mM HDAC1/2 inhibitor (Cayman Chemical Company, BRD6688, 1404562-17-9) for 24 hr, followed by treatment with HDAC1/2i + 2 mg/mL doxycycline for 24 hr. 5 million cells were then washed in PBS and lysed in RIP lysis buffer (50 mM Tris pH 8.0, 100 mM NaCl, 5 mM EDTA, 0.5% NP-40, 1 cOmplete Protease inhibitor tablet) in the presence of RNAse inhibitor (Thermo Scientific, RiboLock RNAse Inhibitor, EO0382). Cells were sonicated on a Covaris (Fill level 10, Duty Cycle 5%, PIP 140 Watts, 200 cycles/burst, 5 × 1 min/sample), and debris was pelleted for 15 min at 21,000xg at 4'C. Cleared supernatant was then added to 100 uL FLAG magnetic beads (Sigma, anti-FLAG M2 Magnetic Beads, M8823). After 2 hr incubation at room temperature, beads were washed three times in cell lysis buffer and resuspended in Trizol. RNA was extracted using the RNEasy mini kit (Qiagen, 74106). qRT-PCR was run on a Roche Lightcycler 480 using the Brilliant II SYBR Green qPCR Master Mix (Stratagene, 600804) using the following primers: Xist-F: 5'GCTGGTTCGTCTATCTTGTGGG3', Xist-R: 5'CAGAGTAGCGAGGACTTGAAGAG3', ActB-F: 5'TCCTAGCACCATGAAGATCAAGATC3', ActB-R: 5'CTGCTTGCTGATCCACATCTG3'.

## Multimapping for repeat-derived reads in ChIP-seq and ATAC-seq data

After finding that SPEN regulates repeat regions, we re-analyzed the ChIP-seq and ATAC-seq data to include repetitive regions, which are usually removed in the standard analysis. We allowed each multimapping read to randomly assign to one place in the genome, without filter reads by mapping quality. The differential peaks is consistent to the results from unique mapped reads. We used the alignment files with multiple mapped reads for all the meta-analysis to avoid losing information.

## Evolutionary analysis of spen

Multiple alignments of the SPEN CDS sequence were download from UCSC, including human, chimpanzee, rhesus, mouse, dog, opossum, platypus, chicken, zebrafish. We separated the CDS of SPEN into three domain regions (RID, SPOC and RRM).

We estimated the number of nonsynonymous substitutions per nonsynonymous site ($dN$), the number of synonymous substitutions per synony- mous site ($dS$), and the $dN$/$dS$ ratio, for each region using maximum likelihood as implemented in the codeml program of the PAML software package (*Yang, 2007*; *Kumar et al., 2016*). Model 0 were used to estimate the global evolution rate and model one were used to estimate branch specific evolution rate.

## Integrated analysis of ChIP-seq, RNA-seq, and ATAC-seq data

Differentially expressed genes were identified from RNA-seq data. The 1 kb region around the promoter of differentially expressed genes were used to count the signal from ChIP-seq and ATAC-seq data. The fold change of gene expression, histone modification and chromatin accessibility between WT and *Spen* KO were compared using a R package, named tableplot (*Tennekes et al., 2013*).

The average plots and heatmap from histone modification, RNA expression, accessibility for interested regions were made using R package ngsplot (*Shen et al., 2014*).

## Repeat divergence analysis

The divergence scores were download from repeatMasker through UCSC genome browser. We separated the ETn family by repeats with binding sites detected from irCLIP or not and the divergence score from mismatches (as a percentage) were compared. icSHAPE data analysis.

The icSHAPE data from Sun et al. were downloaded from GEO under accession GSE117840. The sequence data reads were collapsed to remove PCR duplicates and trimmed to remove 3' adapters following the icSHAPE computational pipeline (*Flynn et al., 2016*). The processed reads were

mapped to the expressed repeat sequence using bowtie2 with parameters suggested by *Flynn et al., 2016*. The icSHAPE score were then calculated following the previous description. Repeat sequence with read depth more than 75 were kept for the following analysis. The icSHAPE score of SPEN binding sites and their flanking region(+ / - 20 bp) were extracted, The aggregated icSHAPE activity score were plotted for each base across non-overlapped binding locations on the Ent repeats.

## irCLIP sequencing data processing

Sequencing output reads were processed by bbmap to remove the duplication on fastq level (*Smith et al., 2017*). Remained reads were trimmed off the 3' solexa adapter and against the sequencing quality q20 by cutadapt (version 2.4). Trimmed reads were mapped first to the RNA bio-types with high repetitiveness by bowtie2 (version 2.2.9) to our custom built indexes: rRNAs (rRNAs downloaded from Ensembl NCBI37/mm9 and a full, non-repeat masked mouse rDNA repeat from GenBank accession No.BK000964), snRNAs (from Ensembl NCBI37/mm9), miscRNAs (from Ensembl NCBI37/mm9), tRNAs (from UCSC table browser NCBI37/mm9), retroposed genes (RetroGenes V6) and RepeatMasker (from UCSC table browser NCBI37/mm9). Remained reads were mapped to the mouse genome NCBI37/mm9 by STAR (version 2.7.1a) with junction file generated from mRNAs and lncRNAs of the Genocode NCBI37/mm9 GTF file. Only reads uniquely mapped to the mouse genome were included in the down-stream analysis. Mapping output files were transformed into bam files (samtools/1.9) and bed files (bedtools/2.27.1) for downstream analysis (*Smith et al., 2017*).

## SPEN binding cluster identification

The RBP binding loci as suggested by the irCLIP method, was defined as one nucleotide shift to the 5' end of each mapped read. Each loci was extended five nucleotides both up and downstream into a 11-nt-long local interval. Only intervals overlapped between two biological replicates were considered reliably detected and were included in the downstream analysis. Overlapping intervals were merged into one cluster. Five nucleotides were trimmed from each side of the cluster to shape the final cluster boundary. Cluster length is required to be longer than one nucleotide for downstream analysis.

SPEN binding clusters on mouse genome and on repeat-regions were identified respectively. Cluster annotations were processed against the Genocode NCBI37/mm9 GTF file and RepeatMasker.

For each experiment condition of each cell type, only clusters with quantification tenfold higher than the observation in its corresponding IgG samples, would be included for the downstream analysis.

## Gene/repeat element level quantification of SPEN binding

Quantification of SPEN binding on each expressed gene or expressed repeat element was calculated as the total amount of read in all binding clusters covering the gene/repeat element. The quantifications were then normalized by gene/repeat expression level.

## SPEN binding RNA motif analysis

RNA motif enrichment analysis were process on SPEN binding clusters on mouse genome and on repeat-regions respectively (Homer version 4.10). Strand-specific sequences defined by SPEN irCLIP clusters were searched for short sequences elements 5-mer, 6-mer, 7-mer and 8-mer in each analysis of hypergeometric enrichment calculations. The searching background was set as expressed transcriptome and expressed repeat regions.

## RNA 2D-structure

2D-structures were predicted by the RNAstructure tool (version 6.0.1) with the maximum free energy model and according the icSHAPE results. 2D-structures of MMETn repeat1125276, ETnERV repeat355428 and RLTR13G_2D were process by algorithm AllSub. 2D-structure of Xist repeat A region #4 and #5 were processed with algorithm DuplexFold.

## Cloning of wild-type and mutant spen RRMs 2–4 for in vitro biochemistry

Wild type mouse Spen cDNA was kindly provided in a D-TOPO vector from the Guttman lab (Caltech). Mutant Spen cDNA corresponding to RRMs 2–4 (F442A, K471A, Y479A, F481A, K513A) was ordered as a gBlock from IDT (Coralville, IA). Primers containing SalI and XhoI restriction enzyme cut sites on the 5'-end and 3'-end respectively of the cDNA corresponding to RRMs 2–4 (aa 336–644) were used to PCR amplify both wild type and RRM3 Mt cDNA inserts for ligation into a pET28b vector containing an N-terminal 10xHis-SUMO tag using the Quick Ligation kit (NEB, Ipswich, MA). Plasmids containing the cDNA of interest were verified by Sanger sequencing prior to expression (Quintara Biosciences, San Francisco, CA).

## Expression and purification of wild-type and mutant spen RRMs 2–4 for in vitro biochemistry

BL21(DE3) expression cells were transformed with pET28b vector containing the cDNA of the protein of interest. 1 L cultures of 2XYT media were grown at 37°C for 3 hr to an $OD_{600}$ between 0.6–1.2. Cells were flash-cooled on ice for 10 min, then 1 mM IPTG was added to induce protein expression overnight at 20°C. Cells were harvested, and pellets were frozen and stored at −20°C until purification. All proteins were subject to a 3-column purification protocol; the wild-type buffers were pH 7.5, while the mutant buffers were pH 8.3 due to differences in predicted pI and anticipated solubility requirements. Specifically, one cOmplete EDTA-free EASYpack protease inhibitor tablet (Roche, Basel, Switzerland) was added to each cell pellet corresponding to 1 L of growth. The tablet and frozen cell pellet were resuspended in 80 mL of lysis buffer (750 mM KCl, 750 mM NaCl, 1 M urea, 50 mM Tris, 10% glycerol, 10 mM imidazole, 1% triton X-100). The cell slurry was sonicated on ice with a Misonix Sonicator 3000 (Misonix, Farmingdale, NY) for 4–6 total minutes in pulses of 15 s followed by 35 s of rest. Membranes and other cellular debris were sedimented away from soluble protein on a Beckman J2-21 large floor centrifuge (Beckman Coulter, Brea, CA). Supernatant containing the recombinant protein was incubated with 10 mL of precleared Ni-NTA beads for 10 min at 4°C. Unbound cellular proteins were cleared by gravity flow. Beads were washed twice; once with 150 mL of lysis buffer, and once with 100 mL of modified lysis buffer containing 20 mM imidazole. Protein was then eluted with 50 mL of similar buffer containing 300 mM imidazole. For the washes and elution, beads were resuspended and incubated with the buffer for 10 min prior to gravity flow through. The elution was dialyzed in 4 L of size exclusion column buffer (500 mM urea, 270 mM KCl, 30 mM NaCl, 50 mM Tris, 7% glycerol, 1–2 mM DTT) for about 2 hr at 4°C. 1–2 mL of 1 mg/mL in-house His-ULP1 [Mossessova and Lima, Mol. Cell (2000), 865–876] was then added, and SUMO tag cleavage of recombinant Spen protein and further dialysis continued overnight at 4°C. Protein precipitate, when present, was pelleted or filtered prior to flowing over the second Ni-NTA column (11 mL beads precleared in column buffer). The Spen fragment flowed through the second Ni column while the His tagged protease and SUMO expression tag were retained on the resin. Flowthrough containing the cleaved Spen construct was concentrated to 2 mL for injection onto the Superdex 200 size exclusion column. Size exclusion fractions are checked for the presence of purified protein by SDS-PAGE. Fractions containing purified protein are combined, concentrated to between 100 μM - 1 mM, aliquoted, flash frozen in liquid nitrogen, and stored at −70°C until binding assays are performed.

RNA generation cDNAs of RNAs of interest with a 5' T7 promoter were ordered as gBlocks from IDT (Coralville, IA). Primers flanking the cDNA containing the EcoRI and BamHI cut sites on the 5' and 3' ends respectively were used to PCR amplify the cDNA for restriction enzyme cloning into a pUC19 vector for cDNA amplification. Ligated plasmids were verified using Sanger sequencing (Quintara Biosciences, San Francisco, CA) and utilized for transcription template generation. Standard PCR using a 3' transcription primer was used to generate in vitro transcription template for run-off transcription. For in vitro transcription, 300 nM PCR template was incubated at 37°C for 2–4 hr with 4 mM each rNTP in 24 mM $Mg^{2+}$, 40 mM Tris pH 8.0, 10 mM DTT, 2 mM spermidine, 0.01% Triton X-100, IPPase (>0.01 U/μL), and in-house T7 polymerase (Batey Lab, CU Boulder). Transcription reactions were gel purified on a 5–12% acrylamide 8M urea 1X TBE slab gel. Reaction products were UV shadowed, appropriate sized bands were excised, and RNA was extracted through a 0.5X TE crush and soak (4°C, 1 hr). Gel bits were filtered and RNA was buffer exchanged in 0.5X TE until

the calculated residual urea was less than 10 µM. RNA was concentrated prior to use; purity and concentration were assessed by the $A_{260}$ and the $A_{260}/A_{280}$ ratio. Concentrations were calculated using the extinction coefficient provided for each RNA sequence by the Scripps extinction coefficient calculator (https://www.scripps.edu/researchservices/old/corefac/biopolymercalc2.html).

## FTSC 3′-end labeling of RNA

350 pmol of purified RNA were treated with 20 mM $NaIO_4$ for 20 min in the dark at room temperature. Excess $NaIO_4$ was precipitated using 26 mM KCl (10 min on ice followed by a hard spin). Supernatant containing the RNA was then ethanol precipitated with the aid of 1 µg glycogen. The RNA pellet was washed 1x with 70% ethanol, and then resuspended in 100 µL of freshly made labeling solution (1.5 mM FTSC, 10% DMF, 100 mM NaOAc pH 5.2) and reacted at 37°C in the dark for 1–2 hr. 280 µL of cold 100% ethanol was then added to precipitate the RNA (ice for $\geq$30 min). The labeled RNA pellet was then washed thoroughly at least 4X with 400 µL of 70% ethanol, resuspended in 30 µL of water, and run through a Sephadex MicroSpin G-25 column (GE Healthcare) to remove excess FTSC. Labeled products were qualitatively assessed for purity and labeling efficiency via denaturing gel analysis; the Cy2 filter on the Typhoon imager (GE Healthcare) was used to visualize the attached fluorophore. $A_{260}$ measurements post-labeling were used to determine RNA concentration. Labeling efficiencies typically allowed for the use of RNA in binding assays between 1–3 nM. FTSC-labeled RNA was stored in dark amber tubes at −20°C until use.

## Fluorescence anisotropy (FA) binding assays

Wild-type and mutant Spen binding were performed at pH 7.5 and 8.3 respectively. RNA concentrations were held as low as possible while still obtaining enough fluorescent signal (1–3 nM). Briefly, purified labeled RNA was snap cooled in 1X binding buffer (45 mM KCl, 5 mM NaCl, 50 mM Tris, 5% glycerol) at 2X the final concentration, and likewise protein titrations were performed separately in 1X binding buffer at 2X final concentration. Protein and RNA were then mixed in a 1:1 vol ratio in a 20 µL reaction and were allowed to come to equilibrium at room temperature in the dark for 40–60 min. Flat bottom low flange 384-well black polystyrene plates (Corning, Corning, NY) were used. Perpendicular and parallel fluorescence intensities ($I_\perp, I_\parallel$) were measured using a ClarioStar Plus FP plate reader (BMG Labtech). Anisotropy values were calculated for each protein titration point where anisotropy = $(I_\parallel - I_\perp)/(I_\parallel + 2*I_\perp)$. Protein concentration was plotted versus associated anisotropy, and data were fit to the simplified binding isotherm (anisotropy ~fraction bound=$S*(P/(K_D+P))+O$ with KaleidaGraph where S and O were saturation and offset respectively, and P was the protein concentration.

## $^{32}$P 5′-end labeling of RNA

50 pmol of RNA were treated with calf intestine phosphatase (NEB) for 1 hr at 37°C. RNA was purified via phenol chloroform extraction followed by ethanol precipitation. RNA was resuspended and γ-ATP was added to the 5′ end of the RNA with T4 polynucleotide kinase (NEB) for 30 min at 37°C. Water was added to dilute the RNA to 1 µM, and the reaction was then run through a Sephadex MicroSpin G-25 column (GE Healthcare) to remove excess γ-ATP. $^{32}$P-labeled RNA was stored at −20°C.

## Competition assay

Wild-type Spen and $^{32}$P-labeled A repeat concentrations were kept constant at 10 µM and 1 µM respectively, and the competition was done at 50 mM Tris (pH 7.5), 45 mM KCl, 5 mM NaCl, and 2.5% glycerol. Different amounts of unlabeled competitor RNA were added, and the amounts were relative to the amount of labeled RNA. Briefly, RNAs were snap cooled in RNA buffer (50 mM Tris (pH 7.5), 45 mM KCl, 5 mM NaCl). Labeled RNA was snap cooled at 4X the final concentration, while competitor RNA was snap cooled at 2X the final concentration. Protein was diluted in protein buffer (50 mM Tris (pH 7.5), 45 mM KCl, 5 mM NaCl, 10% glycerol) to 4X the final concentration. 2.5 µL of 4X protein, 2.5 µL of 4X labeled RNA, and 5 µL of 2X competitor RNA were allowed to come to equilibrium at room temperature for 40–60 min. The reactions were then loaded onto a pre-run 5% acrylamide 0.25X TBE mini native gel at 200V. The gel was run at room temperature at 200V for 10–

15 min, dried, then exposed overnight to a phosphoscreen. Phosphoscreen images were taken on the Typhoon (GE Healthcare) at 50 µm resolution.

## List of RNA sequences used in direct binding and competition

A-repeat from Xist (85 nt):
GGAACTCGTTCCTTCTCCTTAGCCCATCGGGGCCATGGATACCTGCTTTTAACCTTTCTCGG
TCCATCGGGACCTCGGATACCTG

SRA RNA (82 nt):
GGAGCAGGTATGTGATGACATCAGCCGACGCCTGGCACTGCTGCAGGAACAGTGGGC
TGGAGGAAAGTTGTCAATACCTGTA

ERV 205 (86 nt):
GAUUCCAGGCAGAACUGUUGAGCAUAGAUAAUUUUCCCCCCUCAGGCCAGCUUUUUC
UUUUUUUAAAUUUUGUUAAUAAAAGGGAG

ERV 509 (114 nt):
GGUUAAAACUGCUAUUGUUCCAUUGACUGCAGCUUGCAGUUUGAUUUCAAAUUUAAGAUC
UUUUAAUUCACCUGUAUACUGUAAUUAAGAUAAUUACAAGAGUAAUCAUCUUAUG

ERV 495 (113 nt):
GAAAUUUCUCUCUGGGCCUUAUAGCAGGAGUACUCUCGUUCCCUUUUGUGUCUUGUCUAA
UGUCCGGUGCACCAAUCUGUUCUCGUGUUCAAUUCAUGUAUGUUCGUGUCCAGU

ERV 411 (109 nt):
GGUACCUUAAAUCCUUGCUCUCACCCAAAAGAUUCAGAGACAAUAUCCUUUUAUUAC
UUAGGGUUUUAGUUUACUACAAAAGUUUCUACAAAAAAUAAAGCUUUUAUAA

ERV 302 (111 nt):
GAUCAGAGUAACUGUCUUGGCUACAUUCUUUUCUCUCGCCACCUAGCCCCUCUUCUCUUC-
CAGGUUUCCAAAAUGCCUUUCCAGGCUAGAACCCAGGUUGUGGUCUGCUGG

## Code availability statement

Analysis methods used for next generation sequencing data in this paper are described in the appropriate sections of the methods. No new software packages were generated for the analysis of data in this paper.

## Acknowledgements

We thank E Heard and A Wutz for reagents and for helpful discussions. We thank F Fazal, MR Corces and members of the Chang lab for assistance and feedback. Supported by NIH RM1-HG007735, R01-HG004361, and Scleroderma Research Foundation (to HYC), R01-GM120347 (to DSW, RTB), and the Hagey Laboratory for Pediatric Regenerative Medicine and R01DEO26730 (to MTL). HYC is an Investigator of the Howard Hughes Medical Institute.

## Additional information

### Competing interests

Howard Y Chang: Reviewing editor, *eLife*, co-founder of Accent Therapeutics, Boundless Bio, and an advisor of 10x Genomics, Arsenal Therapeutics, and Spring Discovery. Anil Mistry: is an employee of Novartis. The other authors declare that no competing interests exist.

### Funding

| Funder | Grant reference number | Author |
| --- | --- | --- |
| National Human Genome Research Institute | HG007735 | Howard Y Chang |
| National Human Genome Research Institute | HG004361 | Howard Y Chang |
| Scleroderma Research Foundation | | Howard Y Chang |

| | | |
|---|---|---|
| Howard Hughes Medical Institute | | Howard Y Chang |
| National Institute of General Medical Sciences | GM120347 | Robert T Batey<br>Deborah S Wuttke |
| National Institute of Dental and Craniofacial Research | DEO26730 | Michael T Longaker |
| Hagey Laboratory for Pediatric Regenerative Medicine | | Michael T Longaker |

The funders had no role in study design, data collection and interpretation, or the decision to submit the work for publication.

## Author contributions

Ava C Carter, Conceptualization, Investigation, Methodology, Writing - original draft, Writing - review and editing; Jin Xu, Data curation, Formal analysis; Meagan Y Nakamoto, Seung-Woo Cho, Investigation, Methodology; Yuning Wei, Software, Formal analysis, Visualization; Brian J Zarnegar, Formal analysis, Validation; Quanming Shi, Ryan C Ransom, Ankit Salhotra, Rui Li, Diana R Dou, Kathryn E Yost, Anil Mistry, Investigation; James P Broughton, Surya D Nagaraja, Methodology; Michael T Longaker, Resources, Supervision; Paul A Khavari, Resources, Writing - review and editing; Robert T Batey, Deborah S Wuttke, Supervision, Funding acquisition, Visualization; Howard Y Chang, Conceptualization, Supervision, Funding acquisition, Investigation, Writing - original draft, Project administration, Writing - review and editing

## Author ORCIDs

Ava C Carter (iD) https://orcid.org/0000-0002-1892-8742
Jin Xu (iD) http://orcid.org/0000-0003-0944-9835
Diana R Dou (iD) http://orcid.org/0000-0001-6269-9636
Howard Y Chang (iD) https://orcid.org/0000-0002-9459-4393

## Decision letter and Author response

Decision letter https://doi.org/10.7554/eLife.54508.sa1
Author response https://doi.org/10.7554/eLife.54508.sa2

# Additional files

## Supplementary files

- Supplementary file 1. irCLIP-seq clusters identified on mRNA, lncRNA, and TEs.
- Transparent reporting form

## Data availability

All raw and processed sequencing data have been deposited in GEO under accession number GSE131413.

The following dataset was generated:

| Author(s) | Year | Dataset title | Dataset URL | Database and Identifier |
|---|---|---|---|---|
| Carter AC, Xu J, Chang HY | 2020 | Spen links RNA-mediated endogenous retrovirus silencing and X chromosome inactivation | http://www.ncbi.nlm.nih.gov/geo/query/acc.cgi?acc=GSE131413 | NCBI Gene Expression Omnibus, GSE131413 |

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
