## [Decision Letter]

Thank you for submitting your article "Spen links RNA-mediated endogenous retrovirus silencing and X chromosome inactivation" for consideration by *eLife*. Your article has been reviewed by three peer reviewers, including Gene W Yeo as the Reviewing Editor and Reviewer #1, and the evaluation has been overseen by James Manley as the Senior Editor. The following individual involved in review of your submission has agreed to reveal their identity: Molly Hammell (Reviewer #3).

The reviewers have discussed the reviews with one another and the Reviewing Editor has drafted this decision to help you prepare a revised submission.

Summary:

Carter et al. demonstrates that SPEN, an RNA binding protein involved in *Xist* interaction, also binds to ERV elements embedded in transcripts. Structurally, ERV elements appear to be similar to the SPEN binding site in *Xist*. The authors provide data that indicate SPEN represses ERV elements in mouse ES cells. Inactivation of SPEN leads to increase in chromatin accessibility, active histone marks and RNA transcription of ERV element-encoded transcripts. The authors identify that RRM3 domain of SPEN is responsible for binding to ERV and insertion of ERV into an A-repeat deficient *Xist* can rescue binding of *Xist* RNA to SPEN, which result in local gene silencing. Overall the paper is an interesting observation that has for the most part rigorous assays that contribute to our understanding of how *Xist* evolved new domains to recruit protein interactors for gene expression control. There are, however, major concerns that need to be addressed.

Major concerns:

1) Can the authors clarify how the chromatin changes observed at ERV genomic loci change upon SPEN depletion in mESCs as there are some inconsistencies. The high expression of the ERV elements in WT ESCs would be expected to reflect open chromatin at these loci. But are these ERVs then repressed normally? The model of RNA recruitment of SPEN to the ERVs can be better explained to account for these inconsistencies. Also, what fraction of the ERVs with altered chromatin marks have associated expressed transcripts and which ones are differentially expressed in SPEN KO cells? Can the authors present the data as standard DEseq-like scatter/volcano plots?

2) The preference for ERVs, and especially ERV-K's is still unclear. The only thing that links ERVK type TEs to each other is the presence of a lysine-tRNA in the primer binding site (PBS) of those ERV internal sequences. So, given that SPEN is not looking for a specific sequence, but a structure. And, given that SPEN doesn't seem to care about the ERV PBS region, it is difficult to understand why it prefers ERVK elements. Can the authors show that these are structurally different than the other TEs that are not bound? Eg., do they lack the single stranded bubbles associated with SPEN preferred shapes? Or, are these really just following what's expressed at this stage?

3) The results from the experiments replacing the A-repeat in *Xist* with the ERV sequence needs more clarification. The RIP-PCR data does not appear to demonstrate significant binding of SPEN to ERV-XIST for clone 1 and the CLIP-PCR data in the supplement does not show significance testing and have differing data between the two *Xist* primer pairs. The results regarding the repression of ERV-XIST needs to be clarified. It seems that HDAC1/2 inhibition has the largest effect on derepression, even though SPEN is thought to function through HDAC3 in XCI. Which HDACs might be involved in SPEN recruitment to ERV-XIST? In that same line of inquiry, the authors should explain some inconsistencies. They report SPEN induces silencing of the *Xist* RNA instead of silencing the *Xist*-bound chromosome. This suggests that there is some information that could be useful in the comparisons between which TEs were directly bound in the Clip-seq data, which of them had chromatin-based alterations, and which induced changes at the RNA level. There should have been one summary figure of the genomics data that shows how consistent these are with each other. Perhaps what happens at a locus depends on where SPEN binds (on the LTR promoter elements vs. internally)? Perhaps the number of SPEN binding sites on the locus influences how those transcripts are regulated (as they seem to be hinting at in Figure 6E)? Even if this can't explain why SPEN seems to regulate TEs differently from how it contributes to XCI, these comparisons should be explored.

---

## [Author Response]

Major concerns:1) Can the authors clarify how the chromatin changes observed at ERV genomic loci change upon SPEN depletion in mESCs as there are some inconsistencies. The high expression of the ERV elements in WT ESCs would be expected to reflect open chromatin at these loci. But are these ERVs then repressed normally? The model of RNA recruitment of SPEN to the ERVs can be better explained to account for these inconsistencies. Also, what fraction of the ERVs with altered chromatin marks have associated expressed transcripts and which ones are differentially expressed in SPEN KO cells? Can the authors present the data as standard DEseq-like scatter/volcano plots?

The reviewers make a great point, that the relationship between changes in chromatin accessibility or modifications and RNA expression can be better presented. In general, there is low but detectable level of ERV RNA expression in wild type cells. In *Spen* KO cells, some ERV loci gain chromatin access, lose heterochromatic marks, and have elevated RNA expression. These results are consistent with a Spen-mediated surveillance of ERV loci via RNA. We now present this data in Figure 2—figure supplement 1A-F. In panels a-c, we show that there are family-specific changes in chromatin accessibility, H2K9me3 modification, and expression of specific subfamilies of transposable element as determined by DESeq2 in *Spen* KO mESCs. In panels d-f, we show the changes in RNA expression and chromatin modification at the sites that gain accessibility in *Spen* KO mESCs. Consistent with the data presented in Figure 2, we demonstrate that sites gaining accessibility in *Spen* KO mESCs show increased RNA expression, increased enhancer-associated marks (H3K27Ac), and decreased repressive marks (H3K9me3).

2) The preference for ERVs, and especially ERV-K's is still unclear. The only thing that links ERVK type TEs to each other is the presence of a lysine-tRNA in the primer binding site (PBS) of those ERV internal sequences. So, given that SPEN is not looking for a specific sequence, but a structure. And, given that SPEN doesn't seem to care about the ERV PBS region, it is difficult to understand why it prefers ERVK elements. Can the authors show that these are structurally different than the other TEs that are not bound? Eg., do they lack the single stranded bubbles associated with SPEN preferred shapes? Or, are these really just following what's expressed at this stage?

The reviewers raise a good point, that Spen’s apparent preference for ERVKs cannot be explained entirely by their belonging to the ERVK family of TEs. We believe that Spen’s specificity for this subset of ERVK subfamilies has at least two components: (i) the developmental stage during ES cell differentiation, and (ii) the structure of the recognition site on the RNA. First, we observed that Spen does not bind to nor repress all ERVKs. Rather, only specific subsets of ERVK families are targeted, notably the ETn family and RLTR9 and 13 (Figure 2C and Figure 2—figure supplement 1). These subsets of ERVKs have the potential to be expressed in the mESC state, and thus are not as heavily repressed by DNA methylation and other chromatin modifications as other TE families. Thus, they may display some aberrant transcription in mESCs, leading Spen to recognize their RNA products and recruit silencing machinery to silence the locus. Consistent with this idea, we see ectopic occupancy of Oct4, the pluripotency transcription factor on these subsets of ERVKs (Figure 2A, 2B).

Second, within each of these TE subfamilies we find that the elements with the strongest RRM binding in *Spen* KO mESCs show the highest expression level (Author response image 1). Despite these TE families being highly expressed, there are other TE families that show high expression but low Spen binding, suggesting that structural features in TE RNAs also contribute to Spen’s ability to recognize them (Author response image 1). It is difficult to assess the structural features of TEs that are not bound by Spen. For Spen-bound TEs, we know the specific Spen binding site from our irCLIP-seq data and can use the sequence in that specific region to assess RNA structure from icSHAPE and RNA folding models. For unbound TEs, the entire length of the ERVK RNA contains many secondary structures (predicted), so it is difficult to assess which regions are appropriate to compare to the Spen binding sites on bound RNAs. Nonetheless, the Spen irCLIP-seq data show that the RRM binding is highly focal within ETnERV and Xist RNAs (Figure 4), which correspond to specific RNA secondary structures with a GC-rich dsRNA regions interspersed with single-stranded RNA bases. This specificity is consistent with the known RNA binding specificity of Spen RRM (Arieti et al., 2014; Lu et al., 2016).

**Author response image 1. respfig1:** Relationship between expression and Spen RRM binding for ERV RNAs. (**a**) Scatter plot showing the number of RT stops from irCLIP-seq vs. the expression from RNA-seq for all expressed MMETn-int elements. (**b**) Same as in a for all ETnERV elements. (**c**) Table displaying the ERV elements that are expressed in WT or Spen KO mESCs. Columns display the number of expressed elements, the number bound by Spen, the percentage bound by Spen, and the expression rank amongst all ERV families.

3) The results from the experiments replacing the A-repeat in Xist with the ERV sequence needs more clarification. The RIP-PCR data does not appear to demonstrate significant binding of SPEN to ERV-XIST for clone 1 and the CLIP-PCR data in the supplement does not show significance testing and have differing data between the two Xist primer pairs. The results regarding the repression of ERV-XIST needs to be clarified. It seems that HDAC1/2 inhibition has the largest effect on derepression, even though SPEN is thought to function through HDAC3 in XCI. Which HDACs might be involved in SPEN recruitment to ERV-XIST? In that same line of inquiry, the authors should explain some inconsistencies. They report SPEN induces silencing of the Xist RNA instead of silencing the Xist-bound chromosome. This suggests that there is some information that could be useful in the comparisons between which TEs were directly bound in the Clip-seq data, which of them had chromatin-based alterations, and which induced changes at the RNA level. There should have been one summary figure of the genomics data that shows how consistent these are with each other. Perhaps what happens at a locus depends on where SPEN binds (on the LTR promoter elements vs. internally)? Perhaps the number of SPEN binding sites on the locus influences how those transcripts are regulated (as they seem to be hinting at in Figure 6E)? Even if this can't explain why SPEN seems to regulate TEs differently from how it contributes to XCI, these comparisons should be explored.

We apologize for the confusion regarding the Xist-ERV experiments and hope to clarify here and in the main manuscript text. First, we observed that expression of *Xist-ERV* was significantly reduced compared to *Xist-WT* or *Xist-*△*A* upon doxycycline treatment and we found that this failure to induce the chimeric *Xist* RNA was due to epigenetic silencing and not genetic changes in the Tet operator in our cells (Figure 6—figure supplement 2D-G). Recent work from Edith Heard’s lab (Zylicz et al., 2019) showed that HDAC3 inhibition partially blocks X chromosome inactivation. Interestingly, they also found that HDAC1/2 inhibition leads to upregulation of transgenic *Xist* through epigenetic derepression. Thus, we tested whether HDAC1/2i or HDAC3i could relieve Spen-mediated silencing of *Xist-ERV*. We found that both HDAC1/2 and HDAC3 relieved Spen-mediated *Xist* repression, and chose to perform functional experiments in the presence of HDAC1/2i, since we knew this would allow expression of our chimeric *Xist* presumably without affecting the ability of this lncRNA to silence the X chromosome which we wished to test. Our data also indicate that Spen and HDAC3 are responsible for *Xist-ERV* mediated gene repression, as the repression is relieved by knock down of Spen and inhibition of HDAC3 (Figure 6D).

We also wish to clarify the results of the Spen RRM RIP experiments with Xist-ERV clone 1. Even in the presence of HDACi, *Xist-ERV* expression was significantly lower than the expression of *Xist-WT*, making the RIP-qPCR results for this cell line significantly noisier than those obtained with Xist-WT. The results are plotted as the percent of input RNA, and in the case where the input RNA is lowly expressed, this value is inherently noisier. While the result is not statistically significant, this is driven in large part by two outliers (one high and one low, of 6 total). Thus the results for Xist-ERV clone 1 suggest that Spen RRM binds to Xist-ERV and the results from clone 2 show that Spen RRM binding to Xist-ERV is significant.

In Author response image 2 and in Figure 6B, we now present the results of the Spen RRM RIP-qPCR experiments excluding the highest and lowest replicates for each sample, including all controls. Due to the low levels of *Xist* expression, the noise in the assay warrants the removal of these outliers, as the experiment is operating at the lower bound of dynamic range. From this analysis, we conclude that the results in Xist-ERV clones 1 and 2 show that the Spen RRMs bind to *Xist-WT* and *Xist-ERV*, but not to *Xist-ΔA*.

**Author response image 2. respfig2:** Spen RRM RIP-qPCR results with and without outlier correction (related to Figure 6b). (**a**) RIP-qPCR for Xist and ActB in Xist-WT, Xist-ΔA, and Xist-ERV clones 1 and 2 mESCs. qRT-PCR values are normalized to input samples. Error bars represent standard error for 4 independent biological replicates. The highest and lowest values for each sample were removed due to noise. P-values are derived from two-tailed t-test. (**b**) RIP-qPCR for Xist and ActB in Xist-WT, Xist-ΔA, and Xist-ERV clones 1 and 2 mESCs. qRT-PCR values are normalized to input samples. Error bars represent standard error for 6 independent biological replicates. No values were excluded from the analysis. P-values are derived from two-tailed t-test.

We also understand the confusion regarding the CLIP-qPCR presented in the supplemental figure. The reason that these panels did not have significance indicated is that they were performed with technical but not biological replicates. We opted to perform more experiments (n=6 biological replicates) for the RIP-qPCR instead of the CLIP-qPCR, so we have chosen to remove the CLIP-qPCR results from the manuscript. We don’t believe this affects the results presented in Figure 6.

Regarding the mechanism by which Spen silences target loci, we believe that our results are consistent with a model in which Spen recognizes and binds certain transcripts, including ERV-derived transcripts, recruits silencing machinery, including HDACs, and silences the promoter. We provide the list of altered loci with each assay in Figure 2—figure supplement 1. This model is consistent with our results at ERVs as well as our results in the context of the *Xist-ERV* chimeric RNA.

The matter of the overlap between ERVKs bound by Spen RRMs (irCLIP-seq) and the ERVKs that become de-repressed at the level of chromatin modifications and RNA expression is complex. We find consistently that the sites that change at the chromatin and RNA level belong to specific subsets of ERVKs including ETns, and we found that these ERVK subfamilies are also bound by Spen RRMs in multiple cell lines and conditions (Spen FL and RRM-FLAG). However, we find very little overlap between the specific genomic loci that change in accessibility and produce RRM-bound transcripts (1/288 sites that gain accessibility). We believe that this is likely due to differences in when these experiments were performed and the passage of the cells used. For the irCLIP-seq experiments, cells had to be expanded, infected with RRM-FLAG viruses and subcloned. Thus, if Spen’s recognition of ETns and other ERVKs is consistent, but the source of these aberrant expressed ETns in the genome is dynamic, the results are consistent with what we observe. Unfortunately, for technical reasons, these experiments were performed multiple passages apart.

A fascinating mystery of the X inactivation field is how *Xist* itself can be expressed from the inactive X chromosome when this lncRNA is recruiting multiple powerful gene silencing complexes. This duality of potent expression on the inactive X while silencing the remainder of the chromosome is an evolved function of the modern lncRNA. In the context of wild type *Xist*, Spen is able to bind to *Xist* and spread across the chromosome in order to silence distal gene promoters. We found that *Xist-ERV*, our model of a proto-Xist lncRNA, mainly executes cis silencing of its own locus. This implies that additional features are missing in *Xist-ERV*. We agree with the reviewer that the number or quality of RNASpen interaction may determine the difference between local vs. distant gene silencing. It is also worth recalling that *Xist* A-repeat interacts with m6A modification machinery (WTAP, RBM15) as well as RNF20 (Chu et al., 2015; Patil et al., 2016). Some of these interactions are likely missing in *Xist-ERV*. Very recently it was suggested that *Xist* A-repeat has transcriptional anti-terminator activity (Lee et al., 2020), that allows *Xist* RNA to be transcribed in the face of regulatory factors that normally terminate transcription. We believe that solving this longstanding mystery in the field is beyond the scope of the present work, and respectfully suggest our work has clarified the distinct functions that may be gained by a lncRNA when it exapts an endogenous retroviral element.